# Chemical Nature of Metals and Metal-Based Materials in Inactivation of Viruses

**DOI:** 10.3390/nano12142345

**Published:** 2022-07-08

**Authors:** Haozhong Tian, Bin He, Yongguang Yin, Lihong Liu, Jianbo Shi, Ligang Hu, Guibin Jiang

**Affiliations:** 1State Key Laboratory of Environmental Chemistry and Ecotoxicology, Research Center for Eco-Environmental Sciences, Chinese Academy of Sciences, 18 Shuangqing Road, Beijing 100085, China; hztian_st@rcees.ac.cn (H.T.); bhe@rcees.ac.cn (B.H.); ygyin@rcees.ac.cn (Y.Y.); lhliu@rcees.ac.cn (L.L.); jbshi@rcees.ac.cn (J.S.); gbjiang@rcees.ac.cn (G.J.); 2College of Resources and Environment, University of Chinese Academy of Sciences, Beijing 100049, China; 3School of Environment, Hangzhou Institute for Advanced Study, University of Chinese Academy of Sciences, Hangzhou 310024, China; 4School of Environment and Health, Jianghan University, Wuhan 430056, China

**Keywords:** nanoparticles, metallic materials, antiviral, chemical nature, mechanism

## Abstract

In response to the enormous threat to human survival and development caused by the large number of viruses, it is necessary to strengthen the defense against and elimination of viruses. Metallic materials have been used against viruses for thousands of years due to their broad-spectrum antiviral properties, wide sources and excellent physicochemical properties; in particular, metal nanoparticles have advanced biomedical research. However, researchers in different fields hold dissimilar views on the antiviral mechanisms, which has slowed down the antiviral application of metal nanoparticles. As such, this review begins with an exhaustive compilation of previously published work on the antiviral capacity of metal nanoparticles and other materials. Afterwards, the discussion is centered on the antiviral mechanisms of metal nanoparticles at the biological and physicochemical levels. Emphasis is placed on the fact that the strong reducibility of metal nanoparticles may be the main reason for their efficient inactivation of viruses. We hope that this review will benefit the promotion of metal nanoparticles in the antiviral field and expedite the construction of a barrier between humans and viruses.

## 1. Introduction

For decades, human survival and the global order have been severely threatened by viruses, such as novel coronavirus disease 2019 (COVID-19), Ebola virus (EBoV), severe acute respiratory syndrome coronavirus (SARS-CoV) and human immunodeficiency virus (HIV), etc. [1,2,3,4]. Even more alarming is global warming, which shortens viral latency and accelerates the rate of vector-borne infections [5]. Furthermore, the time-consuming process of new drugs discovery and frequent human mobility have impaired human control of viruses [6,7]. Therefore, we should strive to eliminate viruses at the source and establish an effective defense system.

Over a long time, enormous contributions have been made to virus inactivation by many conventional disinfection techniques. For instance, physical techniques consist of ultraviolet (UV) radiation, heating, reverse osmosis, filtration and high pressure [8,9,10]. Chemical techniques consist of ozone oxidation, chlorination, acid, alkali and oxidant methods [11,12,13]. Up to now, these conventional disinfection techniques are still widely used in medical treatment, drinking water purification, sewage treatment and other fields. However, their disadvantages encompassing residual disinfection byproducts, high energy consumption and high operating costs no longer satisfy the requirements of sustainable development [14,15]. Considering the time-consuming process of new drugs discovery and shortcomings of conventional techniques, many researchers have been committed to seeking effective, simple and accessible methods to inhibit virus spread, including artificial intelligence approaches [16].

As a novel technology that has flourished for decades, the unique properties of nanomaterials may provide an alternative solution for controlling the transmission of viruses [17,18,19]. Among the numerous nanomaterials, metal nanoparticles have both the ability of metals to inactivate pathogens and the structural properties of nanoparticles, so an increasing number of functionalized metal nanoparticles have been reported as potential candidates for viral propagation inhibition [20,21,22,23,24,25]. Although the antiviral applications of metal nanoparticles have been summarized in excellent reviews [23,26,27,28,29,30,31,32,33], fewer studies have summarized the antiviral mechanisms of metal nanoparticles. Moreover, researchers in different fields hold dissimilar views on the antiviral mechanisms, which has slowed down the antiviral application of metal nanoparticles.

In reality, the chemical nature of the metal elements may determine the antiviral properties rather than the morphological characteristics of the nanoparticles. To this end, this review begins with an exhaustive compilation of the antiviral performances of metal nanoparticles and other metallic materials. Afterwards, we comprehensively evaluate the antiviral heterogeneity of metallic materials according to the chemical nature of metal elements. Together, we hope that this review will help researchers in various fields to select suitable substrates for antiviral materials based on the chemical nature of metal elements, which will promote the development of novel powerful weapons for virus elimination.

## 2. Antiviral Performances of Different Metallic Materials

Thus far, metal mineral resources are still abundant in the crust of the earth, and humans have been exploiting metal ore for tens of thousands of years on earth [34]. Meanwhile, the recycling and utilization of solid wastes from cars and electronics have ensured a stable supply of metallic materials in recent decades [35]. Besides the relatively low cost of metal mineral extraction and processing, some metallic materials exhibit redox, photocatalytic and structural stability and other characteristics [36,37,38]. Hence, metallic materials have great application potential in defending against and inactivating viruses that spread through various vectors.

The performances and mechanisms of metallic materials as antibacterial agents have been exhaustively summarized [31,39,40,41]. The main mechanisms of bacterial inactivation by metals are as follows: (i) metal binds to the cell wall through electrostatic interactions, destroys the cell wall and causes cytoplasmic efflux; (ii) metal accumulates in the cell membrane of bacteria and damages it, thereby causing increased cell permeability; (iii) metal enters the bacterial cell bound to enzymes and disrupts intracellular metabolism; and (iv) metal induces free radical production in the presence of light, which damages the genetic material of bacteria and hinders bacterial propagation (Figure 1; all figures were created with Adobe Illustrator 2020) [40,42,43,44,45]. Nevertheless, in contrast to the structure and composition of bacteria, viruses have no cell wall, cell membrane, cytoplasm and nucleus, and they have only capsid and genetic material (RNA or DNA) [46]. Hence, the interaction mechanism between virus and metal may be different from that of bacteria, along with a possible decrease in inactivation efficiency, which requires a more comprehensive summary about the antiviral applications of metallic materials.

To date, metallic materials with proven antiviral capability are mainly composed of noble metal elements (e.g., gold and silver) and transition metal elements (e.g., titanium, iron, nickel, copper and zinc) [22,23,26,47,48,49]. These metallic materials can effectively inactivate a variety of viruses, mainly divided into plant viruses (e.g., cucumber mosaic virus), animal viruses (e.g., influenza virus, adenovirus, norovirus and HIV) and bacteriophages [22,23,50,51,52,53]; these viruses are shown in Figure 2.

In previous studies, a variety of antiviral efficiencies and mechanisms were shown to exist due to the different conditions comprising forms of metallic materials and test environments, which confounds the judgment on the critical mechanism of virus inactivation by metals. Hence, in order to assess whether the chemical nature of the metal element or the morphology of the material is the major contributor to the antiviral effect, we provide a synopsis of the antiviral performances and mechanisms of metal nanoparticles, metal ions, pure metals and alloys and metal compounds.

### 2.1. Metal Nanoparticles

Nanotechnology is an emerging field that has flourished in recent decades, and a large number of materials with nanostructures have been synthesized by this technique, such as nanoparticles (NPs), nanowires, nanorods and nanofilms [17,18,19]. Among these nanomaterials, functional NPs with a large specific surface area, high surface reactivity, good biocompatibility and optical and electronic properties have been systematically studied in biomedical fields such as drug delivery, biosensors, disease detection and therapy (Figure 3) [27,54]. Among all NPs, metal NPs have both the ability of metals to inactivate pathogens and the structural properties of NPs, so an increasing number of functionalized metal NPs have been reported as highly effective inhibitors of viral proliferation [20,21,22,23,24,25].

#### 2.1.1. Copper Nanoparticles

Copper iodide (CuI) is an active inorganic catalyst with excellent photodegradation ability [55], and it has also attracted attention from some researchers for its antiviral aspects [56]. An earlier investigation first reported the inactivation effect of CuI nanoparticles on the 2009 pandemic influenza virus (H1N1) [20]. The results showed that the amount of viruses with a concentration of 10^6^ PFU/mL (plaque assay) decreased 3.5 log_10_ dramatically after 60 min of exposure to CuI NPs. The median particle diameter (D50) of CuI NPs was 160 nm, and the concentration for 50% of the maximal effect (EC50) was approximately 17 μg/mL.

Meanwhile, it was found that the destruction of viral proteins (e.g., hemagglutinin and neuraminidase) by CuI might be the main cause of viral inactivation. Hemagglutinin is a necessary protein for the virus to enter host cells through endocytosis [57], and neuraminidase is an essential protein for the virus to release from the surface of host cells [58]. There are two substances that can destroy viral proteins, namely, hydroxyl radicals (·OH) and superoxide ions (O_2_^•−^), which are generated by the reductant Cu^+^ released from CuI in water. The reaction is described in Equations (1)–(3) [59]. Moreover, the superimposition of ·OH and O_2_^•−^ can cause more extensive chain scission of protein molecules.
(1)$${\mathrm{Cu}}^{+}+O_{2}(\mathrm{aq})\to {\mathrm{Cu}}^{2+}+O_{2}{}^{•-}$$
(2)$$2O_{2}{}^{•-}+2H^{+}\to H_{2}O_{2}+O_{2}$$
(3)$${\mathrm{Cu}}^{+}+H_{2}O_{2}\to {\mathrm{Cu}}^{2+}+{\mathrm{OH}}^{-}+\cdot \mathrm{OH}$$

More recently, the removal efficiency of bacteriophage MS2 by copper oxide nanoparticles (Cu_x_O_y_ NPs, the particle size is 30–50 nm) with different valences was tested [60]. When the initial concentration of viruses was 10^4^ PFU/mL, CuO had no effect on the concentration of viruses in the water, while the concentration of viruses in the water dropped by 3 log_10_ after filtration through Cu_2_O or Cu materials. Moreover, it was argued that the highly efficient removal of bacteriophage MS2 by Cu_x_O_y_ NPs was due to the efficient adsorption of negatively charged viruses by materials with surface positive charges. That is, electrostatic interactions play a major role in virus inactivation.

#### 2.1.2. Silver Nanoparticles

Silver nanoparticles (Ag NPs) are considered to be a promising antibacterial and antiviral material because of their high specific surface area, oxidation resistance and electrical conductivity [26]. An earlier study first discovered the size-dependent interaction of Ag NPs with HIV-1 [21]. In this study, 10^5^ TCID_50_/mL (50% tissue culture infectious doses) of HIV-1 was treated with 3–21 nm Ag NPs, and the infectivity of HIV-1 was reduced to an undetectable level after 3 h of exposure to silver NPs at concentrations higher than 25 μg/mL.

HIV-1 has a lipid envelope and its surface-attached glycoprotein gp120 can bind to the CD4 receptor on host cells, which is a key step in HIV infection of host cells (Figure 4A) [61,62,63]. That is, gp120 glycoprotein is a potential target for the binding of Ag NPs. If the Ag NPs can bind to the gp120 glycoprotein and destroy it, the interaction between the virus and the host cell will be terminated. To this end, the study also explored the role of Ag NPs and gp120 glycoprotein [21]. The results showed that the diameter of Ag NPs attached to the virus envelope was not more than 10 nm, and the arrangement spacing on the envelope surface was approximately equal to the distribution spacing between gp120 glycoproteins. It was very certain that the combination of Ag NPs with sulfur residues in the gp120 glycoproteins of HIV-1 leads to virus inactivation (Figure 4B).

The size-dependent antiviral properties have also been confirmed in other studies. For instance, a research team earlier evaluated the inhibition of severe acute respiratory syndrome coronavirus 2 (SARS-CoV-2) by Ag NPs with various sizes [64]. In this study, Ag NPs with 2–15 nm size showed robust inhibition of SARS-CoV-2, and the viral RNA copies (1.3 × 10^9^/mL) decreased by 1.41 log_10_ after 1 h of treatment with 10 nm Ag NPs. Combined with previous studies on viral proteins, it can be inferred that the disulfide bonds on the spike protein and angiotensin-converting enzyme-2 receptors may be essential targets for Ag NPs to attack SARS-CoV-2.

A recent study also investigated the virucidal activity of Ag NPs with different particle sizes and surface modifications [65]. In this study, 10^5.35^ TCID_50_/mL SARS-CoV-2 was treated with 5, 20, 50 and 100 nm Ag NPs, and it was found that the virucidal effects of Ag NPs further diminished with increasing particle size. In particular, after 24 h of treatment with 5 nm polyvinyl pyrrolidone (PVP)-coated Ag NPs, the titer of SARS-CoV-2 significantly reduced by 1.65 log_10_. Ag NPs with smaller particle size exhibit larger specific surface area, implying greater enhancement of their ability to bind to viruses. Moreover, this study confirmed that the zeta potential of Ag NPs was positively correlated with antiviral efficacy, which provided a new insight for further research on the antiviral mechanism of metallic nanoparticles.

With the intensive research on Ag NPs, products for controlling pathogen transmission have been developed. A research team investigated the resistance of Ag NPs modified silica hybrid composites (Ag30-SiO_2_) to bacteriophage MS2 and murine norovirus (MNV) [50]. In this study, approximately 10^6^ PFU/mL of bacteriophage MS2 or MNV was exposed to 400 nm Ag30-SiO_2_, and Ag30-SiO_2_ can effectively inactivate bacteriophage MS2 and MNV in deionized water, tap water, surface water and groundwater. Compared with bacteriophage MS2, MNV was more sensitive to Ag NPs, and the titer of MNV was reduced by more than 3 log_10_ only after 1 h of exposure to Ag30-SiO_2_.

However, the antiviral mechanism of Ag30-SiO_2_ was not explored in this study. It was only inferred from the antibacterial mechanism of Ag NPs that virus inactivation might be caused by the interaction of thiol groups with Ag^+^, and Ag^+^ was produced from the dissolution of Ag30-SiO_2_ in water. Neither bacteriophage MS2 nor MNV has an envelope, which indirectly indicates that the different sensitivities of these two viruses to Ag30-SiO_2_ may be due to the difference in the content of their thiol groups.

More recently, a novel mask decorated with Ag NPs has been fabricated to inhibit the transmission of human coronavirus [66]. In this study, Ag NPs coating was created on glass, face masks, and cotton textiles by reactive blade-coating technology, and the average diameter of Ag NPs was approximately 27 nm. Within 30 min of virus contact with the Ag NPs coated glass, mask and cotton, the titer of human coronavirus 229E (10^7.05^ PFU/mL) dropped by 3.72 log_10_, 3.51 log_10_ and 3.15 log_10_, respectively. These findings demonstrate once again the strong virucidal effect of Ag NPs, and further advance the use of Ag NPs in personal protective equipment.

#### 2.1.3. Nickel Nanoparticles

Nickel has excellent properties such as high-temperature stability, strength and corrosion resistance, and it is one of the metals widely used in many fields [36]. In fact, nickel-containing compounds have been shown to significantly inhibit bacterial growth in many previous studies [67,68], while there are few reports on nickel as a tool to control viral transmission. Recent studies have reported the effect of nickel oxide nanostructures (NONS) on the control of cucumber mosaic virus (CMV) [22]. In this study, NONS with sizes ranging from 15 to 20 nm and cucumber plants infected with CMV were treated with NONS by foliar spraying and soil wetting. Compared to the non-treated cucumber plants, the NONS-treated cucumber plants effectively avoided CMV accumulation and infection.

The results showed that NONS induced the expression of defense-related genes after entering the cells of cucumber plants, thus making the cells resistant to CMV. In addition, another possible mechanism of viral inactivation is that NONS with photocatalytic activity can induce the production of reactive oxygen species (ROS, ·OH and O_2_^•−^). ROS can promote the peroxidation of phospholipid components, which leads to severe disruption of viral function and eventual inactivation [69,70].

One additional study explored the inactivation effect of nanoscale bimetallic Ni/Fe NPs on bacteriophage f2 [71]. In this study, the average size of the Ni/Fe NPs was 92.6 ± 3.5 nm. The results showed that Ni NPs had no obvious inactivation effect on the virus, while Ni/Fe NPs (Fe:Ni = 3:1) could inactivate all bacteriophages f2 (4 × 10^6^ PFU/mL) in solution within 30 min, and the inactivation efficiency was significantly better than that of Fe NPs. As the ratio of Fe increased, the virus removal efficiency of Ni/Fe NPs first increased but then decreased, and the inactivation efficiency of bacteriophage f2 by Ni/Fe NPs was highest at Fe:Ni = 5:1. Therefore, Ni may play a catalytic role in the virus inactivation process. Further studies on the mechanism confirmed that virus inactivation was mainly caused by the production of ·OH and O_2_^•−^ during the process of Fe oxidation. These two ROS had a potent lethal effect on the virus, and Ni accelerated the oxidation reaction of Fe.

#### 2.1.4. Gold Nanoparticles

Gold nanoparticles (Au NPs) exhibit excellent biocompatibility and low toxicity, which make great contributions to pathogen inhibition and disease treatment [28]. Currently, many research teams have reported the inhibitory effect and mechanism of Au NPs on viruses. For instance, an earlier investigation reported the inhibitory effect of Au NPs on vesicular stomatitis virus (VSV), and researchers found different viral inhibition mechanisms dependent on particle size [23]. VSV is an enveloped negative-sense RNA virus that generally infects a variety of mammals including swine and horses and causes acute diseases in these animals. This virus also infects humans, usually resulting in mild influenza syndrome or asymptomatic infection [72]. The study results showed that Au NPs with a particle size greater than or equal to 52 nm effectively inhibited 60–70% of virus binding to erythrocytes, while Au NPs with a particle size of 19 nm inhibited only 18% of virus binding to erythrocytes. Au NPs with a size equal to or larger than VSV can form VSV-Au NP clusters with the virus, while Au NPs with a size smaller than VSV can only limitedly reduce the binding sites between virus and host cells. Consequently, Au NPs could inhibit virus infection by preventing VSV binding to host cells, and this inhibition was largely positively correlated with the particle size of Au NPs.

Another study evaluated the antiviral efficacy of Au NPs against herpes simplex virus (HSV) [73]. HSV is an enveloped double-stranded DNA virus that can cause infections of the labials, oculars or genitals, and might damage human neurons [74]. In this study, Au NPs with a particle size of 18.27 nm effectively inhibited HSV-1 and HSV-2 (10^4^ PFU/mL) from infecting Vero cells. The toxicity of Au NPs on Vero cells was lower than that of acyclovir, a drug used extensively in the clinic to control HSV infection. It should be noted that toxicological testing with cells does not account for longer-term toxicity that may be seen in in vivo models such as mice. Moreover, the study explored the mechanism by which Au NPs inhibit HSV infection. On the one hand, Au NPs might directly attach to the surface of HSV to eliminate the infectivity of the virus. On the other hand, Au NPs might be first absorbed by host cells, and then inhibit infection by interfering with viral proliferation.

Measles is a highly contagious and potentially fatal disease caused by measles virus (MeV) [75]. Although measles is still endemic in many countries, safe and effective vaccines have led to a significant reduction in morbidity. One more piece of good news is that researchers have confirmed the potent virucidal effect of Au NPs against MeV [76]. In this study, Au NPs with a size of approximately 11 nm were synthesized using garlic extract as reducing agent, and the titer of MeV (3 × 10^5^ PFU/mL) dropped by 0.8 log_10_ after mixing with Au NPs for 3 h. Moreover, they confirmed that Au NPs directly attach to the measles virus envelope, blocking the union with host cells.

More recently, a research team evaluated the virucidal effect of porous Au NPs against influenza virus [77]. In this study, porous Au NPs with a size of approximately 150 nm displayed antiviral activity on various virus strains, such as H1N1, H3N2 and H9N2. After the 1 h treatment with Au NPs, the titer of H1N1 (10^6^ EID_50_/mL) decreased by 0.6 log_10_. Furthermore, it was found that Au NPs have higher affinity to the disulfide bonds in hemagglutinin. As previously mentioned [78], disulfide bonds play a crucial role in membrane fusion. Thus, to block the spread of the virus, the disulfide bonds in hemagglutinin may be an effective target.

#### 2.1.5. Iron Nanoparticles

Nano zero-valent iron (NZVI) has high surface reactivity and reducibility, and it has been applied to remove environmental pollutants such as poorly biodegradable organics, nitrates, etc. [79]. In recent years, researchers have found that NZVI is a novel material with great potential in inactivating pathogenic microorganisms [80]. In one published study, the inactivation effect of NZVI (size ≈ 200 nm) on bacteriophage MS2 was reported [81]. The results of the study showed that NZVI inactivation of phage MS2 was divided into two stages. The first was the rapid stage, and the concentration of bacteriophage MS2 (10^7^ PFU/mL) decreased by 4 log_10_ during 0–5 min of co-culture. The second was the slow stage, and the concentration of bacteriophage MS2 decreased by 3 log_10_ during 5–240 min of co-culture. Meanwhile, the researchers found that O_2_^•−^ played a major role in the first stage of bacteriophage MS2 inactivation, and ·OH played a major role in the second stage. Moreover, the inactivation efficiency of NZVI against bacteriophage MS2 was positively correlated with its specific surface area.

A similar study reported the optimal inactivation effect of NZVI (size < 100 nm) on bacteriophage f2 in water [82]. In this study, the dose of NZVI and the rotation rate of the constant temperature incubator had a significant effect on the removal rate of bacteriophage f2. When the solution pH = 5.12, rotation rate = 148.75 rpm, NZVI dosage = 49.07 mg/L, and virus concentration = 3.5 × 10^6^ PFU/mL, NZVI can inactivate the virus by 5.51 log_10_.

More recently, the inactivation mechanism of NZVI (size ≈ 50 nm) on bacteriophage f2 was demonstrated in another study by the same research team [24]. In this study, the inactivation efficiency of bacteriophage f2 was positively correlated with the dosage of NZVI. When the initial concentration of viruses was 10^6^ PFU/mL, the virus removal efficiency of 0.5 mmol NZVI after 60 min reaction under anaerobic and aerobic conditions was 2.4 and 4.1 log_10_, respectively, which proved that oxygen had a significant contribution to bacteriophage f2. Under aerobic conditions, Fe^2+^ and O_2_ generated O_2_^•−^ and H_2_O_2_ via electron transfer and then generated ·OH via the Fenton reaction. These two ROS were the main contributors to the removal of bacteriophage f2, and the reaction is described in Equations (4)–(6). In addition, some NZVI was oxidized to generate Fe_3_O_4_ or Fe_2_O_3_, and these two iron oxides could adsorb the virus to their surface and then inactivate the virus (Figure 5). This study also proved that bacteriophage f2 inactivation was divided into two stages. In the early stage of the reaction, NZVI inhibited the infectivity of the virus, and then the virus was inactivated by ROS generated from the environment in the later stage. The antiviral performances of metal nanoparticles are summarized in Table 1.
(4)$${\mathrm{Fe}}^{2+}+O_{2}\to {\mathrm{Fe}}^{3+}+O_{2}{}^{•-}$$
(5)$${\mathrm{Fe}}^{2+}+O_{2}{}^{•-}+2H^{+}\to {\mathrm{Fe}}^{3+}+H_{2}O_{2}$$
(6)$${\mathrm{Fe}}^{2+}+H_{2}O_{2}\to {\mathrm{Fe}}^{3+}+\cdot \mathrm{OH}+{\mathrm{OH}}^{-}$$

### 2.2. Metal Ions

The morphology of the metal is one of the key factors affecting its ability to inactivate viruses [83]. Although the antibacterial performances of solid metallic materials have been confirmed, some metallic materials may not have a significant inactivation effect on viruses in solution [51]. This is because some metals do not have high solubility and the limited amount of metal ions released from the material surface may affect the virus inactivation efficiency [53]. Consequently, some researchers have co-cultured metal salts with viruses in an aqueous environment to clarify the inactivation mechanism and whether there is concentration dependence.

#### 2.2.1. Copper Ions

Copper ions (Cu^2+^) are one of the most commonly used antibacterial agents in recent centuries, and copper has become a more promising metal for virus inactivation due to its relatively low cost and toxicity [84]. The antiviral effects of Cu^2+^ and Zn^2+^ on the avian influenza virus H9N2 were reported by previous researchers [85], which provided a reference for formulating new programs to prevent the spread of the virus. In this study, Zn^2+^ had no inactivating effect on the avian influenza virus, while Cu^2+^ at a concentration of 25 μM/L could reduce the titer of H9N2 viruses (10^6^ TCID_50_/mL) by nearly 4 log_10_ within 6 h. The amount of inactivated viruses is proportional to the exposure time of Cu^2+^, and the inactivation rate is proportional to the concentration of Cu^2+^. Additionally, researchers found that the activities of viral hemagglutinin (HA) and neuraminidase (NA) did not decrease after Cu^2+^ addition, while the morphology of the viruses changed abnormally. Therefore, the inactivation mechanism of Cu^2+^ on H9N2 may be related to structural damage. Further studies are needed to determine whether the structural changes of the virus affect the normal function of its RNA.

Another study reported the effect of cotton textiles synthesized from Cu^2+^ containing zeolite (Cu-Zeo) on the inactivation of three H5 subtype influenza viruses (H5N1-C, H5N1-S and H5N3) [53]. The results of this study showed that Cu^2+^ contained in the zeolite could rapidly and effectively inactivate three H5 subtypes of influenza viruses (4.3 log_10_ reduction of H5N1 virus within 10 min), and different strains of the virus had different sensitivities to Cu^2+^. In contrast, no decrease in viral titer was observed on cotton textiles synthesized from zeolite without Cu^2+^. No profound disruption to the viral gene by Cu^2+^ was observed in this study, and the activity of the viral hemagglutinin protein was not affected by Cu^2+^. Similarly, the inactivation of influenza virus may be related to the destruction of virus structure by Cu^2+^. Based on the great adsorption and regeneration ability of zeolite [86], the fabrics synthesized by Cu-Zeo can be used as protective clothing for hospitals or farms in the future to prevent airborne viruses from harming human health.

#### 2.2.2. Silver Ions

In line with copper ions, silver ions (Ag^+^) have also been one of the most commonly used antibacterial agents in recent centuries [84]. Ag^+^ can not only combine with acid radicals and halogens, but can also form coordination compounds with amines, carboxylic acids, thiols and other substances [87]. Accordingly, there are theoretically multiple Ag^+^ attack targets on the surface of the virus. Once Ag^+^ binds to these sites, the virus may be effectively inactivated. Recent studies reported the inactivation effects of two silver compounds on pathogenic influenza A virus and bacteriophage Qβ, namely AgNO_3_ (high water solubility) and Ag_2_O (medium water solubility) [88]. In this experiment, the titer of influenza A virus (10^8^ TCID_50_/mL) significantly decreased by 6 log_10_ after exposure to AgNO_3_ and Ag_2_O for 30 min, while the titer of bacteriophage Qβ decreased by only 3 and 2 log_10_ after exposure to AgNO_3_ and Ag_2_O for 60 min, respectively. These results showed that the inactivation efficiency depends on the solubility of silver compounds, and Ag^+^ has higher inactivation efficiency for viruses without an envelope. In addition, the researchers also found that Ag^+^ could break the disulfide and thiol bonds of proteins, and the reaction is described in Equations (7) and (8) [89]. Thus, the viral protein will denature when exposed to Ag^+^, and viruses lose their ability to infect the host cell and eventually die.
(7)$${\mathrm{Ag}}^{+}+R-S-S-R\to 2R-S-\mathrm{Ag}$$
(8)$${\mathrm{Ag}}^{+}+R-\mathrm{SH}\to H^{+}+R-S-\mathrm{Ag}$$

In another study, the ability of Ag^+^ to resist the sacbrood virus (SBV, a widely spread virus that resides in bees) was first investigated [90]. In this study, apiaries affected by SBV were selected as targets for evaluating antiviral effects. Ag^+^ was randomly added to the brown sugar syrup ([Ag^+^] = 0.2 mg/L) in one apiary as the test group. No Ag^+^ was added to the brown sugar syrup of another apiary as a control group. The results showed that the population density and activity of the bees in the test group remained well within 30 days of observation, and the bees survived longer. However, the population density of bees in the control group decreased rapidly after 8 days of observation, the worker bees stopped their daily work, and many bees even escaped from the hives. These results demonstrated that Ag^+^ is therapeutic but not curative for bees infected with SBV.

#### 2.2.3. Zinc Ions

Zinc is an essential trace element in organisms, that not only participates in cell division and energy metabolism [91,92], but also exhibits excellent antibacterial and antiviral performances [93,94]. The performance of two zinc salts against transmissible gastroenteritis virus (TGEV) was tested by previous researchers [95]. TGEV is an enveloped single-stranded RNA coronavirus that is highly contagious in swine and can cause severe enteritis with high mortality [96]. The results of this study indicated that zinc ions (Zn^2+^) had no direct inactivation ability prior to TGEV infection and could not prevent the virus from binding to host cells. However, the titer of TGEV (10^6.7^ TCID_50_/mL) significantly decreased by 1.4 log_10_ after the addition of Zn^2+^. Zn^2+^ mediated antiviral effects by inhibiting viral penetration or reducing the life cycle of the virus inside host cells, and researchers speculated that the intracellular target of Zn^2+^ might be viral RNA polymerase. In addition, the virus inactivation efficiency was positively correlated with the concentration of Zn^2+^, but Zn^2+^ could cause cytotoxic effects at high concentrations.

More recently, the inhibitory effect of nitroporphyrin-zinc complexes on HIV-1 and macaque simian immunodeficiency virus (SIVmac) was demonstrated in another investigation [52]. In this study, it was found that the Zn^2+^ doped porphyrin complex had stronger virus suppression ability than the porphyrin complex not doped with Zn^2+^. Moreover, the nitroporphyrin-zinc complex was not specific for virus inhibition, that is, it could effectively inhibit both HIV-1 and SIVmac. Researchers found that porphyrins might enter the host cell membrane and inhibit fusion between the virus and the cell membrane, thereby hindering viral spread (Figure 6). However, this study did not clarify the strengthening mechanism of virus suppression by doped Zn^2+^ in the material.

#### 2.2.4. Others

Besides the study of single ions, other research also demonstrated the antiviral properties of hybrid coatings synthesized from silver, copper and zinc cations against HIV-1, influenza virus H1N1, human coxsackievirus B3, human herpes virus 1 and dengue virus [97]. The study results showed that the titer of HIV-1 (1.66 × 10^5^ TCID_50_/mL) decreased by 3.1 log_10_ after 20 min of exposure on the coating. The inactivation rate of other viruses on the coating was slower, for instance, the titers of influenza (3.63 × 10^5^ TCID_50_/mL), dengue (3.98 × 10^6^ TCID_50_/mL) and herpesviruses (1.00 × 10^6^ TCID_50_/mL) decreased by 1.3 log_10_, 2.3 log_10_ and 5.0 log_10_, respectively, after 240 min of exposure to the coating. In particular, the titer of the unenveloped coxsackievirus B3 did not change significantly.

In reality, a major contribution to the inactivation of these viruses was the release of metal ions from the hybrid coating. Cu^2+^ and Ag^+^ killed the virus by directly cleaving the envelope or binding to the thiol group of the protein, while the exact mechanism of virus inactivation by Zn^2+^ remains unknown. Furthermore, the research team also found that the activity of mammalian cells did not change significantly even after four hours of exposure. In the future, metal ion hybrid coatings can be applied in various devices in the medical field to block the spread of pathogenic microorganisms on solid surfaces. The antiviral performances of metal ions are summarized in Table 2.

### 2.3. Pure Metals and Alloys

Pure metals and alloys are widely distributed and abundantly stored in the crust of the earth. They can be industrialized and commercialized without processing or light processing [34,98]. Furthermore, pure metal and alloy materials also have excellent physical and chemical properties, such as structural stability, ductility and corrosion resistance [99,100]. Thus, the direct application of pure metals or alloys to inhibit virus spread will avoid energy consumption and environmental pollution caused by the processing of materials.

Previous studies have shown that the virus could not only spread through water and air, but could also adhere to the surface of solid materials and survive for a period of time [101]. When the hands of susceptible people touch these solid surfaces with pathogens, and then touch their eyes, nose or mouth, the risk of pandemic spread is undoubtedly increased [101]. To this end, researchers have been investigating whether solid surfaces covered with metal coatings could effectively reduce the contact transmission of various pathogens (e.g., bacteria, viruses and fungi) [51,102]. It is certain that the research in this field will provide a reference for applications of antiviral metal coatings on solid surfaces with high human contact frequency, such as scalpels, door handles and stair handrails.

#### 2.3.1. Copper and Copper Alloys

The application of copper in disease treatment can be traced back to the 19th century or even earlier. In the past two decades, some studies have proven that different types of viruses, bacteria and fungi can be quickly killed on the surface of copper [39]. For instance, a research team selected nursing homes as experimental targets to explore the cut-off effect of copper on virus transmission [47]. In this study, door handles, handrails and grab-bars in half of the area were replaced with copper, and no adjustments were made in the other half to compare the number of patients infected with the virus. The results of this study showed that the copper surface could reduce the risk of healthcare-associated infections (HAIs), but only for keratoconjunctivitis (caused by adenovirus) and gastroenteritis (caused by norovirus), two pathogens which spread by hand. For airborne viruses (e.g., influenza A virus), copper surfaces did not exhibit protective effects.

Moreover, another research team compared the effects of copper and stainless steel surfaces on the number of influenza A viruses [51]. In this study, the number of viruses (2 × 10^6^ virus particles/mL) only decreased by 0.3 log_10_ after 6 h of incubation on the stainless steel surface, and decreased by 0.6 log_10_ after 24 h of incubation. However, after 30 min of incubation on the copper surface, the number of viruses dropped by 0.6 log_10_, and the number of viruses dropped by 4 log_10_ after incubation for 6 h. Otherwise, the researchers speculated that copper might prevent virus replication by damaging viral RNA.

More recently, the efficacy of a series of copper alloys to inactivate human coronavirus (HCoV-229E) was determined through experiments [103]. In this study, approximately 5 × 10^4^ PFU/mL of HCoV-229E was stained with copper, copper-nickel alloys (containing 90% copper) and brass (containing 70% copper) in the range of 1 cm^2^, respectively. The results showed that the virus inactivation rate was directly proportional to the percentage of copper in the material. Almost all coronaviruses can be inactivated (3 log_10_ reduction) within 30 min of exposure to 90% copper alloys. Even if exposed to alloys with low copper content (70%), the coronavirus can be completely inactivated (3 log_10_ reduction) within 60 min. In particular, copper caused the fragmentation of the viral genome during inactivation, thus ensuring that inactivation was irreversible. Moreover, the researchers found that the Cu^2+^ and Cu^+^ dissolved from the copper alloy played a major role in virus inactivation, and O_2_^•−^ enhanced the inactivation effect. However, they did not clarify which component of the virus was destroyed during the inactivation process by copper ions.

#### 2.3.2. Iron and Iron Alloys

Zero-valent iron (ZVI) is an active metal with strong reducibility that has been widely used in wastewater treatment, drinking water purification and groundwater remediation [104]. In addition, ZVI has also been explored for its ability to inactivate viruses. For instance, the efficacy of ZVI against Aichi virus, Adenovirus 41, bacteriophage ΦX174 and MS2 in water was investigated in an earlier study [105]. Aichi virus is a non-enveloped human enteric virus that can infect humans through water and food and then cause diarrhea, vomiting, fever and other symptoms [106]. Adenovirus is also a non-enveloped virus that infects humans, typically leading to respiratory infections, gastroenteritis and conjunctivitis [107]. In this study, ZVI could effectively remove indicator virus (10^5^ PFU/mL) within 9–10 min of contact time, and the removal efficiency was 4.5–6 log_10_. In addition, the researchers also found that the virus removal efficiency of ZVI was positively correlated with the virus concentration in the water. The antiviral performances of pure metals and alloys are summarized in Table 3.

### 2.4. Metal Compounds

Metal compounds (e.g., copper oxide and iron oxide) are also widely distributed in nature, and these metal compounds have thermal stability, chemical stability, photocatalysis and other characteristics [37,108]. Although many previous studies have proven that solid metal compounds have excellent antibacterial ability, studies on the virus inactivation by metal compounds are scarce [109,110].

#### 2.4.1. Copper Compounds

Copper oxide (Cu_2_O) and cuprous oxide (CuO) are the two oxide forms of copper, and the inactivation effects of these two compounds on influenza A virus and bacteriophage Qβ have been reported by researchers [88]. In this study, after 30 min of incubation with Cu_2_O particles, the titer of influenza A virus (10^8^ TCID_50_/mL) dropped sharply by 3.7 log_10_. In contrast, after 30 min of contact with CuO particles, the titer of influenza A virus did not decrease significantly. Similar to influenza A virus, after 30 min of incubation with Cu_2_O particles, the titer of bacteriophage (10^9.5^ TCID_50_/mL) Qβ decreased by 5.8 log_10_, while CuO hardly affected the titer of bacteriophage Qβ. Although influenza A virus has a viral envelope and bacteriophage has no envelope [88], this does not affect the antiviral capacity of copper compounds.

In addition, the researchers found that Cu_2_O could greatly reduce the titers of hemagglutinin (HA) and neuraminidase (NA), while the titers of HA and NA were basically unchanged during incubation with CuO. HA is a necessary protein for the virus to enter host cells through endocytosis [57], and NA is an essential protein for the virus to be released from the surface of host cells [58]. Therefore, when viruses are exposed to Cu_2_O, the functions of HA and NA proteins are damaged, and then the viruses lose the ability to infect host cells and are ultimately inactivated.

A similar study reported the antiviral activity of solid copper compounds in multiple oxidation states [111]. In this experiment, the bacteriophage Qβ titer (2.5 × 10^9^ PFU/mL) could drop by 4 to 6 log_10_ within 30 min of exposure on the surface of cuprous compounds (Cu_2_O, Cu_2_S and CuI), while exposure on the surface of cupric compounds (CuO and CuS) resulted in little change in the virus titer. Apparently, cuprous compounds have more effective antiviral properties than cupric compounds. In order to explore the antiviral mechanism, researchers conducted experiments to verify three substances that may exert antiviral effects, namely, ROS, leached copper and solid-state compounds (Figure 7).

It is certain that ROS has the ability to destroy microbial proteins and nucleic acids [112]. However, in this study, ROS was not a major factor responsible for virus inactivation. Moreover, it was found that Cu^2+^ and Cu^+^ did not significantly reduce the titer of bacteriophage Qβ; this was contrary to the results of previous studies, which demonstrated that Cu^2+^ could effectively inactivate viruses by binding to proteins and nucleic acids [53,85]. Ultimately, the researchers believed that the direct contact of the virus with solid copper compounds made a major contribution to virus inactivation, and the underlying cause of inactivation may be that Cu_2_O has a higher adsorption capacity for viral proteins.

#### 2.4.2. Iron Compounds

Iron oxide is also a metal oxide widely distributed in nature, and the catalytic activity of iron oxides will be significantly enhanced under irradiation with sunlight [37]. This unique photoreaction property induces the generation of ROS, which provides a greater possibility for the inactivation of viruses [113]. The inactivation effect of oxide-coated sand (IOCS) on the bacteriophage MS2 and ΦX174 (human indicator virus) has been reported by previous researchers [114]. In this study, the iron oxide coating in the dark significantly enhanced the adsorption capacity of sand on virus but did not cause virus inactivation, and IOCS could desorb infectious viruses according to changes in solution conditions. However, IOCS could significantly adsorb bacteriophage MS2 and caused its inactivation under sunlight irradiation.

Another study reported the effect of iron oxide (Fe_2_O_3_) ceramic membranes on virus removal in water [115]. In this study, Fe_2_O_3_ rapidly inactivated the bacteriophage P22 (10^7^ PFU/mL) within 7 h (2.0 log_10_ reduction), and then the inactivation rate gradually decreased with the extension of contact time. Therefore, the inactivation process was mainly divided into two stages. In the fast stage, the active sites on the material surface were sufficient to bind the virus rapidly. In the slower stage, the active sites on the Fe_2_O_3_ surface were close to saturation, and the virus needed to diffuse to the active sites inside the material. In addition, the binding between Fe_2_O_3_ and the virus was mainly dependent on electrostatic interactions, and Fe_2_O_3_ could maintain strong adhesion with bacteriophage P22 when the pH was 4–6.

More recently, the inactivation effect of iron oxide on viruses in wastewater has been confirmed [116]. In this study, wüstite (a mineral form of FeO) and bacteriophage MS2 (10^6^ PFU/mL) were exposed to solar radiation for the test. The results showed that wüstite had very fast inactivation reaction kinetics, which reduced the concentration of viruses by 5 log_10_ within 30 min. Moreover, researchers have explored the virus inactivation mechanism. On the one hand, iron oxide can directly bind with viruses through electrostatic interactions to achieve inactivation (Figure 8A). On the other hand, under the catalysis of light, a part of the dissolved iron generated ROS or other oxidants through redox reactions, and these substances can directly damage the virus capsid or envelope (Figure 8B).

#### 2.4.3. Titanium Compounds

Titanium dioxide (TiO_2_) is a cheap and highly stable catalyst that can activate its photocatalytic oxidation ability under sunlight or ultraviolet radiation [117]. Under light radiation, the electrons of TiO_2_ are excited from the valance band (${\mathrm{hv}}_{\mathrm{vb}}^{+}$) to the conduction band ($e_{\mathrm{cb}}^{-}$) and electron-hole pairs are generated (Figure 9) [118]. Meanwhile, molecular oxygen can be rapidly reduced by electrons to generate superoxide ions. On the other hand, the hole can directly react with water to form hydroxide radicals [119,120]. The reaction is described in Equations (9)–(12). In particular, ROS has the ability to effectively destroy the proteins and nucleic acids of pathogenic microorganisms [112], so that many studies have focused on the inactivation effect of TiO_2_ on viruses.
(9)$${\mathrm{TiO}}_{2}+\mathrm{hv}\to e_{\mathrm{cb}}^{-}+{\mathrm{hv}}_{\mathrm{vb}}^{+}$$
(10)$$e_{\mathrm{cb}}^{-}+O_{2}\to O_{2}{}^{•-}$$
(11)$${\mathrm{hv}}_{\mathrm{vb}}^{+}+{\mathrm{OH}}^{-}\to \cdot \mathrm{OH}$$
(12)$${\mathrm{hv}}_{\mathrm{vb}}^{+}+H_{2}O\to \cdot \mathrm{OH}+H^{+}$$

For instance, the antiviral activity of TiO_2_-modified hydroxyapatite composites (HA/TiO_2_) was reported in an earlier study [49]. When the viruses were exposed to HA/TiO_2_ (0.5 mg/mL) and ultraviolet irradiation for 60 min, the H1N1 titer (2.6 × 10^7^ PFU/mL) was significantly reduced by approximately 3 log_10_. HA or TiO_2_ alone did not show a significant elimination effect on the virus. The researchers inferred that H1N1 virus was first adsorbed on the surface of the material by HA, and then TiO_2_ was activated by ultraviolet light to produce ROS (e.g., ·OH and O_2_^•−^), which damaged the viral envelope.

Recently, other researchers reported the antiviral performance of Cu-TiO_2_ nanofibers in a single virus system and a virus/bacteria mixed system [121]. The research results showed that in a single virus system, when the visible light intensity was 100 mW/cm^2^, the initial virus concentration was 10^5^ PFU/mL, and the Cu-TiO_2_ concentration was 75 mg/L, the concentration of bacteriophage f2 rapidly decreased by 5 log_10_ within 120 min. However, in the mixed system of virus and *E. coli*, the inactivation effect of Cu-TiO_2_ on bacteriophage f2 decreased significantly. Within a certain range, the removal efficiency of the virus was positively correlated with light intensity, temperature and the dosage of catalyst. The removal efficiency of the virus was negatively correlated with the initial concentration of the virus. Meanwhile, the researchers also found that the substances leading to the inactivation of bacteriophage f2 were mainly ·OH, comprising those bound on the surface of the photocatalyst and free in the bulk phase. In fact, the free ROS played a more important role. The antiviral performances of the metal compounds are summarized in Table 4.

## 3. Antiviral Mechanisms at Biological Level

Up to now, researchers have not reached a consensus on the antiviral mechanisms of metal nanoparticles, and some studies are still at the hypothetical stage. In some previous reviews, the focus was only on the antiviral properties of a metal element [30,32,33], or the performances of a class of materials (e.g., nanoparticles) applied in virus inactivation [25,29,122]. The antiviral mechanisms of various metallic materials have been summarized in fewer reviews. From the above cited literature, we found that there are various antiviral forms of metal nanoparticles at the biological level. Since metal ions are inevitably released from the surface of nanoparticles, it is difficult to distinguish whether nanoparticles or metal ions can effectively inactivate viruses.

Therefore, at the biological level, we tentatively divide the complex antiviral mechanisms into two categories based on the observed alterations in viral morphology, structure, genetic material and other phenomena. Ⅰ. Blockade of virus spread and infection, which may still retain viral activity. Ⅱ. Direct inactivation of the virus results in loss of viral activity and replicative capacity. A more detailed description of the antiviral mechanisms by metal is provided below, and these mechanisms are illustrated in Figure 10.


I.Blockade of virus spread and infection(1)Porous metallic materials or metallic materials with positive charges on the surface can effectively remove viruses through physical adsorption. The virus is only transferred between different phases, and it remains active under certain conditions and can also be released from the surface of the material.(2)Metal nanoparticles or metal ions can bind to the membrane of the host cell, or attach to the surface of the viral envelope or capsid. Both types of binding inhibit fusion between the virus and the host cell membrane, thereby hindering the spread and infection of the virus.(3)When the metallic material enters the host cell, the expression of its viral defense-related genes is activated so that the cell develops resistance to the virus, which can also inhibit the spread and infection of the virus.II.Direct inactivation of the virus(1)Metallic materials can directly destroy the envelope of the virus after contact with the virus or bind to the glycoprotein on the surface of the virus envelope, resulting in virus inactivation.(2)Metallic materials can directly damage the genetic material (DNA or RNA) of the virus and prevent the virus from replicating.(3)Metallic materials can cleave disulfide and thiol bonds of proteins (e.g., hemagglutinin, neuraminidase and RNA polymerase) in the virus, thereby preventing virus replication and inhibiting virus spread and infection. Hemagglutinin is a necessary protein for the virus to enter host cells through endocytosis, and neuraminidase is an essential protein for the virus to be released from the surface of host cells.(4)Metallic materials can react with oxidants and reductants in the environment to generate reactive oxygen species (e.g., hydroxyl radicals and superoxide anions), which can effectively damage the proteins and genetic material of viruses.(5)Metallic materials with photocatalytic activity can induce the generation of reactive oxygen species and other oxidants, and then these oxidants promote the peroxidation of phospholipids, resulting in severe destruction of viral functions.(6)Metallic materials without photocatalytic activity can also act as catalysts for virus inactivation and accelerate the rate of virus inactivation.


## 4. Potential Antiviral Mechanisms at Physicochemical Level

Due to different test environments and types of metallic materials and viruses, diverse antiviral mechanisms continue to emerge. Researchers in the fields of biology and medicine have paid more attention to antiviral mechanisms at the biomacromolecular level [29,45,123,124], and those in the chemical and materials science fields have paid more attention to antiviral mechanisms at the physicochemical level [25,32,42,44]. They have barely dissected the underlying reasons leading to efficient virus inactivation by metal nanoparticles, which has slowed down the antiviral application of metal nanoparticles. In fact, we speculate that the highly effective antiviral effect of metal nanoparticles depends on the chemical nature of the element.

For porous solid metallic materials that interact with viruses through adsorption, the structural characteristics are the main factors affecting the adsorption capacity, while the types of metal elements have little effect on physical adsorption. In general, a large specific surface area, high porosity and dispersion can increase the chance of contact between viruses and adsorption sites on metallic materials [125], as well as increase the adsorption capacity for viruses. In addition, for solid metallic materials that interact with viruses through chemical bonds, the excellent structural features increase the probability of chemical reaction between the virus and the active sites on the surface of the material, and improve the inactivation efficiency of the material against viruses.

Apparently, adsorption just alters the spatial location of the virus but has limited efficacy for virus inactivation. From the discussion on the biological antiviral mechanisms of metal nanoparticles in the previous section, we know that achieving the radical inactivation of viruses mainly relies on the material releasing a large number of metal ions (M^n+^) that can bind to the virus, and the material-induced redox reaction that can damage the virus. Thus, in order to clarify the difference in the antiviral efficiency of metals, we analyzed other physicochemical properties of various metal elements, including the chelation reaction equilibrium constant, hydrate ion radius, ionic potential and standard electrode potential.

### 4.1. Chelation Reaction Equilibrium Constant

Clearly, M^n+^ is not only used as an essential cofactor to catalyze and regulate; it can also drive cytotoxicity reactions in bacteria [126]. During the past several decades, researchers have confirmed that many proteins in bacteria bind to M^n+^ through coordination chemistry, such as transmembrane transporters, chaperones for diffusion within the cytoplasm and metal regulatory sensors. These proteins account for approximately one-third of all structural characteristic proteins [127]. Likewise, a variety of structural and non-structural proteins are also distributed within the virus, such as membrane proteins, channel proteins and proteins involved in transcription, etc. [128]. The functional groups (e.g., amino, carboxyl, hydroxyl, etc.) of amino acid residues in these proteins have multiple binding sites to chelate with M^n+^ (Figure 11A) [129,130,131]. Separately, the negatively charged DNA and RNA strands also provide numerous sites to interact with M^n+^ through electrostatic or coordination (Figure 11B) [132,133].

When metal nanoparticles are mixed with viruses, a portion of the M^n+^ released into the environment undergoes chelation reactions with viruses. The chelation constant is commonly used to assess the stability of complexes, but the composition and structure of viral proteins, DNA and RNA are different, causing difficulties in determining and contrasting the chelation constants between viruses and M^n+^. Therefore, we select the commonly used chelating agent EDTA instead of biomacromolecules, and employ the chelation reaction between EDTA and M^n+^ as a reference for the interaction between viruses and M^n+^. The reaction is described in Equation (13). The reason for this is that EDTA has similar components to biomacromolecules, such as nitrogen atoms and carboxyl groups, and most of the chelation equilibrium constants of M^n+^ with EDTA can be found in the JESS database (Appendix A) [134]. Then, the antiviral rates of M^n+^ were calculated by Equation (14); the data used for calculations were obtained from experimental results in the cited literature. All detailed data were collated in Appendix A.
(13)$$M^{n+}+Y^{4-}\to {\mathrm{MY}}^{(n-4)+}$$
where: M^n+^ is metal ions; Y^4−^ is EDTA^4−^.
(14)$$V=\frac{\mathrm{lg}(N_{0}/N_{t})}{[M^{n+}]\times t}$$
where: V is the antiviral rate of M^n+^, (mM^−1^·h^−1^); N_0_ is the titer of virus in the solution at the initial time, (PFU/mL or TCID_50_/mL); N_t_ is the titer of virus in the solution at t time, (PFU/mL or TCID_50_/mL); [M^n+^] is the concentration of M^n+^ in the solution, (mM); t is the incubation time of M^n+^ and virus, h.

In general, the equilibrium constant of the chelation reaction has a positive correlation with the stability of the complex. The data in Appendix A show that alkali M^n+^ (e.g., Li^+^, Na^+^ and K^+^) cannot form stable complexes with EDTA, while alkaline earth M^n+^ and transition state M^n+^ can form stable complexes with EDTA. These results indirectly prove that many M^n+^ have the potential to combine with viral biomacromolecules through chelation reactions. However, there is no significant correlation between the equilibrium constant of the chelation reaction and the antiviral rate of M^n+^, and few studies have shown that M^n+^ such as Cd^2+^ (lg K_0_ = 18.24) and Ca^2+^ (lg K_0_ = 12.39) have significant antiviral properties (Figure 12).

In reality, M^n+^ with high affinity to viral biomacromolecules is not necessarily toxic to viruses, for many M^n+^ are attached to the active site of viral proteins to support enzymatic activity. For instance, an earlier study observed chelated manganese ions on catalytic residues of the NS5B RNA-dependent RNA polymerase of hepatitis C virus [135], and another study observed chelated zinc ions on catalytic residues of the nsp RNA-dependent RNA polymerase of COVID-19 virus [136]. When the concentration of a certain M^n+^ increases around the virus, it instead promotes viral protein synthesis or enhances enzyme activity. Therefore, the chelation of M^n+^ with viral biomacromolecules does not necessarily lead to the inactivation of the virus, and the chelation equilibrium constant may not well explain the difference in antiviral ability among metallic materials. Moreover, if M^n+^ is responsible for virus inactivation by chelation chemistry, we consider that one of two conditions must be met. First, the concentration of M^n+^ is high enough to exceed the threshold of virus tolerance, and second, M^n+^ has a stronger affinity for viral biomacromolecules that can displace other M^n+^ from the binding site of the original protein, thereby altering the structure and function of the viral protein.

### 4.2. Hydrate Ion Radius

For intact cells, there are many ion pumps and ion channels on their biological membranes that control the transmembrane transport of inorganic ions. When the virus infects the host cell, it will encode the synthesis of viroporins. Viroporins are small hydrophobic transmembrane proteins that oligomerize to form selective ion channels in the membrane of host cells [137]. Furthermore, viroporins alter the permeability of the cell membrane, allowing Na^+^, K^+^ and Ca^2+^ to cross the host cell membrane. Thus, if we apply M^n+^ to treat already infected host cells or to inactivate viruses that have infected host cells, the degree of difficulty for M^n+^ to enter the host cells also needs to be considered.

On the one hand, the selectivity of viroporins, ion pumps and ion channels on the host cell membrane influences the transmembrane transport of M^n+^. On the other hand, ionic radius is an important factor affecting the uptake of M^n+^ by host cells. In an aqueous environment, M^n+^ generally combines with water molecules, and the hydrated radius is the apparent ionic size rather than the ionic radius [138]. Hydrated ions require dehydration before passing through channels and transporters, and a larger ionic radius means more energy will be consumed in the host cells [139]. For that, cells are more inclined to leave ions with a larger radius on the cell membrane or cell wall [140]. The hydrated radius of some M^n+^ has been collected in previous studies (Appendix A).

At present, the standard values of hydrated radii of M^n+^ are unclear. Due to the different test environments or theoretical methods in various studies, the data listed in Appendix A may be biased. As shown in Figure 13, K^+^, Ag^+^ and Na^+^ have small hydrated radii and easily enter the host cells, but neither K^+^ nor Na^+^ has antiviral ability. Fe^3+^, Cr^3+^ and Al^3+^ have large hydrated radii and easily attach to the cell wall or cell membrane, but Fe^3+^ and Cr^3+^ do not inactivate the virus efficiently. In fact, ion transmembrane transport is also affected by the type of channels and transporters, transmembrane pressure and other factors. Hence, the hydrated radius may not be the main factor that affects the antiviral performance of the material. In brief, for those M^n+^ that can destroy various enzymes and genetic material in the virus, ions with smaller hydrated radii may have greater antiviral capacity. Whereas ions with larger hydrated radii, which may not yet have the opportunity to exert antiviral effects, are entrapped on the host cell membrane or cell wall.

### 4.3. Ionic Potential

Electrostatic force is a particularly important interaction between microorganisms and inorganic particles. On the one hand, the superposition of protonated and non-protonated states of functional groups (e.g., carboxyl and amino groups) on the non-enveloped virus protein coat results in the accumulation of a net charge on the viral surface [141]. On the other hand, there are cylindrical pores on the viral surface that penetrate the entire protein shell, thereby connecting the internal DNA or RNA of the virus to the external medium. In particular, each residue of DNA carries a negatively charged phosphoryl group (Figure 11B), and DNA composed of thousands of nucleotides accumulates a large charge density, as does RNA [142]. Therefore, M^n+^ can not only provide empty orbitals accepting electrons of DNA or RNA to form complexes, but can also bind with viruses through electrostatic interactions [132].

The ionic potential (Φ) is a measure of charge density, which is used to evaluate the ability of ions to electrostatically attract ions with opposite charges and repel with the same charge [138]. The ionic potential is related to the ionic charge (Z) and ionic radius (r), and it can be calculated by Equation (15) [143]. All calculated results are listed in Appendix A.
(15)$$\Phi ={\left(\frac{Z}{r}\right)}^{0.5},\;\Phi =\frac{Z}{r},\;\Phi =\frac{Z^{2}}{r}$$
where: Z is the ionic charge; r is the ionic radius, (nm).

Generally, in the case of a net negative charge on the viral surface, M^n+^ with a larger ionic potential means that it is more electrostatically attractive to the virus. As shown in Figure 14, Cr^3+^ and Fe^3+^ have larger ionic potential and stronger electrostatic attraction with viruses, while Ag^+^ has smaller ionic potential and weaker electrostatic attraction with viruses. In fact, Ag^+^ has excellent antiviral ability, while Cr^3+^ and Fe^3+^ can hardly inactivate viruses effectively. The strong electrostatic interaction force does not represent excellent virus inactivation ability; thus, we conjecture that the ionic potential is also not a determining factor affecting the antiviral ability of metallic materials. For M^n+^ with larger ionic potential, stronger electrostatic attraction may contribute to limiting virus spread and blocking virus invasion into host cells.

In addition, the electrostatic interactions between viruses and metallic materials are also affected by ion concentration, solution pH and coat protein functional groups. For example, viruses exhibit significant aggregation between particles due to weakened electrostatic repulsive interactions at lower pH [144]. Significant aggregation of virus particles will reduce the interaction area between the virus and M^n+^ or metal nanoparticles, which may weaken the antiviral ability of metallic materials. Certainly, increasing the dose of metallic materials can overcome these factors, which affect the antiviral capacity, but this does not make sense as high doses can also cause damage to normal surviving host cells.

### 4.4. Standard Electrode Potential

Apparently, van der Waals forces, coordination reactions and electrostatic forces cause little damage to the virus, and only redox reactions remain that can cause lethal damage to the virus. We have found in some previous studies that metallic materials with different valence states of the same element showed different inactivation efficiencies against viruses, and those with low valence states showed better inactivation effects than those with high valence states (e.g., cuprous (Ⅰ) is superior to copper (Ⅱ), ferrous (Ⅱ) is superior to ferric (Ⅲ)) [24,111]. We speculate that this result may be caused by M^n+^ in a low valence state with stronger reducibility. M^n+^ with stronger reducibility is more likely to react with oxygen in the environment to generate ROS that is lethal to the virus, and the reaction equation is described in Equation (16) (the products are exemplified by O_2_^•−^). In order to verify this conjecture, we queried the previous manuals and literature to find standard electrode potentials for each metal element, which is an important indicator to measure their redox ability (Appendix A) [145].
(16)$$M^{n+}+O_{2}(\mathrm{aq})\to M^{(n+1)+}+O_{2}{}^{•-}$$

Among the different elements, the larger the standard electrode potential of the redox couple, the stronger the oxidation ability of the substance in the oxidized state. Under the same element, metal elements in lower valence states possess stronger reducibility. As shown in Figure 15, the reductive order is Fe > Fe^2+^ > Fe^3+^, Cu^+^ > Cu^2+^, Ag > Ag^+^, Au > Au^3+^, consistent with the order of antiviral rate. Thus, metal nanoparticles or ions with stronger reducibility are more likely to have antiviral ability. Metal nanoparticles or ions that have strong reducibility and in which the outermost electrons do not form a stable structure are suitable as antiviral materials, because they can be used as reducing agents to react with oxygen to generate ROS. Furthermore, although the outermost electrons of Ag^+^ form a stable structure, Ag^+^ has been confirmed to be reduced to Ag NPs under light driving, and Ag NPs could in turn generate ROS to inactivate viruses [146]. This result also indirectly reflects the current insufficient research on the antiviral mechanism of metallic material, especially for the exploration of redox reactions in the process of virus inactivation.

Up to now, numerous studies have proven that major biomolecules such as lipids, carbohydrates, proteins, DNA and RNA can all react with ROS [147]. For bacteria, ROS preferentially attacks the phospholipid bilayer and lipopolysaccharide of the bacterial cell membrane, and oxidizes unsaturated fatty acids on the cell membrane to form lipid-peroxyl radicals [148]. For enveloped viruses, the components of the envelope are similar to those of bacterial cell membranes, both composed of a phospholipid bilayer and glycoproteins [149], so that ROS can also damage the viral envelope. Upon passage through the viral envelope, ROS can significantly interfere with protein function, including altering protein structure, oxidizing amino acids, modifying sulfur groups, and carbonylation [150,151]. In addition, ROS with strong oxidative capacity can also react with the genetic material of viruses. On the one hand, ROS cleaves the phosphodiester bonds between DNA duplexes to turn them into single strands. On the other hand, ROS can also directly damage the bases of guanine and adenine [152]. In contrast to bacteria, viruses are unable to synthesize enzymes that regulate ROS balance, such as superoxide dismutase, glutathione reductase, catalase, etc. [153]. Thus, ROS can significantly inactivate viruses, and redox reactions are lethal to viruses.

In summary, we seem to have found some regularity in the past studies insofar as redox ability is a key factor affecting the antiviral performances of metallic materials. In the future, metal nanoparticles or ions with strong reducibility should be widely involved in the field of antiviral materials research. Although the chelation reaction equilibrium constant, hydrated radius and ionic potential are not major factors, these physicochemical properties may also affect the antiviral mechanism and efficiency of the materials and are also important reference factors in the process of selecting antiviral materials. Thus, metallic materials that have been identified in some studies that could not effectively inactivate viruses may also have antiviral ability; they just did not play a role in a suitable location. For example, if the binding force between the host cell membrane and metallic materials is strong, metallic materials that inactivate viruses through ROS have difficulty in entering cells to exert antiviral effects.

### 4.5. Others

In addition, we also summarize several other points based on the research results in the cited literature above. Generally, the inactivation efficiency of most metal nanoparticles against viruses is directly proportional to the dosage of materials and the time of exposure, but their relationship is not completely linear. The inactivation efficiency of metal nanoparticles on viruses is affected by many factors, such as material type, material size, material dosage, virus type, virus dosage, humidity, solution pH, reaction temperature, etc. Because these contributing factors are complex and variable, as well as the current paucity of mechanistic studies on virus inactivation by metal nanoparticles, virus inactivation may also be a complex process in which multiple mechanisms co-exist. In particular, researchers are still divided on whether the envelope of the virus can affect the inactivation efficiency of metal nanoparticles.

On balance, metallic materials have broad-spectrum antiviral performance to inactivate a variety of viruses, encompassing plant viruses and animal viruses. Moreover, most metallic materials may have lower toxicity and exhibit good biocompatibility. Moreover, we also found that antiviral studies of metal nanoparticles are significantly more prevalent than other metallic materials, and metal nanoparticles have a greater potential for practical applications.

## 5. Challenges and Future Perspectives

Through this review, we can see that metallic materials, especially metal nanoparticles, have gained much attention for antiviral applications over the past decades, and this field will continue to provide exciting challenges and opportunities in the future. For instance, previous studies have proven that smaller nanoparticles had more significant virucidal effects [65]. Paradoxically, smaller nanoparticles result in higher biological toxicity [154,155], which means we need to weigh the virucidal effect and biological toxicity of the nanoparticle. Furthermore, although no secondary pollutants are generated during the inactivation of viruses by metallic materials, metal ions and nanoparticles are still inevitably released into the environment. Therefore, it is important to evaluate the effects and risks of metallic materials on environmental health and safety.

Some previous researchers have revealed the uncertain toxicity of metallic materials and evaluated the long-term chronic effects on non-target organisms exposed to metallic materials. The research results showed that, on the one hand, metal nanoparticles can accumulate significantly in organisms. Once nanoparticles are released into aquatic and soil environments, they will not only be enriched in aquatic sediment and soil, but will also accumulate significantly in algae, fish, clams, plankton, benthos and terrestrial plants [156,157,158,159]. Later on, these released harmful heavy metals are then amplified stepwise through the food chain into higher organisms, which may cause damage to the kidneys, liver and other organs after human ingestion [160]. In addition, metal nanoparticles diffused through aerosols in the air are also easily deposited in human lungs [161].

On the other hand, excessive metallic materials may directly damage the tissues and organs of the organism. For example, heavy metals may affect the diversity of certain non-target microbial communities [162] and inhibit the germination and growth of plants [163]. Meanwhile, excessive metal nanoparticles can reduce the survival probability of fish embryos and affect the development of larval organs (e.g., defective eyes and abnormal gills) in the long term [164,165]. In particular, excessive metal elements in the human body may cause serious damage to the liver, kidneys, intestines, central nervous system and reproductive system [166]. Currently, researchers generally believe that metal toxicity to organisms is due to ROS generated intracellularly or extracellularly that damages the cell structure and reduces cell activity [163,167]. To this end, it is necessary to further evaluate the toxicity and bioaccumulation of metallic materials in various model organisms. It is important not only to clarify the migration and transformation mechanism of metallic materials in the environment, but also to specifically monitor the content of ROS released by metallic materials into the environment during virus inactivation.

There are several other issues that need to be resolved before widely applying metallic materials to defend against and inactivate viruses. Some previous studies have proven that bacteria could gradually adapt to metal nanoparticles and develop resistance to metals [168], while whether the virus could evolve resistance genes to metal materials has not yet reached a unified conclusion. Therefore, further research is needed to evaluate the resistance of the virus to metallic materials. In addition, nanoparticles can form agglomerations through van der Waals forces and electrostatic forces, which greatly limits the virus inactivation ability of nanoparticles [169]. Hence, pretreatment methods need to be implemented to disperse the nanoparticles before inactivating the virus [21,60]. Moreover, some metallic materials only adsorb viruses instead of inactivating them. Viruses that remain active under certain conditions can also be released back into the environment, which may still pose a public health risk [114]. Therefore, for metallic materials that only adsorb the virus, a secondary treatment technique is also required to completely inactivate the virus.

In reality, compared with conventional disinfection technology, we value the potential of metal nanoparticles in recycling. Previous studies have shown that the recovery of magnetic nanoparticles can be achieved by applying an external magnetic field [170]. For typical non-magnetic metal nanoparticles, magnetic elements can also be doped in the material and recycled by magnets [171]. In addition, metallic material disinfection devices can be followed by other treatment processes for the purpose of recovering metals, such as physical and chemical adsorption, chemical precipitation, membrane filtration and ion exchange [172]. If the metallic material can be recycled and reused in the disinfection process, it will not only save the cost of virus inactivation, but also reduce the residue of metal in the environment to avoid metal bioaccumulation.

Currently, the supply of new drugs has slowed down, and the struggle with pathogens has become increasingly acute [173]. In recent years, we have been facing a fierce global pandemic, the constant variability of viruses, and some diseases for which we have not yet developed specific medicines to cure, such as HIV and COVID-19 [174,175]. Hence, it is urgent to develop drugs associated with metallic materials to treat patients infected with viruses or to prevent the virus from entering host cells for replication, and future research should carry out as much pilot and large-scale field research as possible to accelerate the process of industrialization and commercial application.

Finally, is it ever thought that metal resistance to viruses could be the result of viral evolution over hundreds of millions of years? Perhaps the metal is just a certain signaling molecule or a modulator between the virus, the virus and the host. This signaling molecule or regulator may be used to kill competitors and relieve survival stress. More interestingly, not all viruses are deleterious. Oncolytic viruses (such as VSV) infect and kill cancer cells without harming normal cells [176]. Thus, viruses can be appropriately guided with metallic materials as a treatment for other diseases, such as targeted therapy of tumor cells utilizing the metal nanoparticle labeled VSV [177,178].

## 6. Conclusions

Up to now, there have been a large number of studies that have reported on the antiviral performances of metal nanoparticles, including the virus inactivation efficiency, the defense and inactivation mechanisms, and the factors affecting antiviral activity. From the analysis of the chemical nature of metal elements, we conclude that redox ability may be a key factor affecting the antiviral ability, and the chelation reaction equilibrium constant, hydrate ion radius and ionic potential are the secondary factors. We hope that this conclusion will help researchers in various fields to select suitable substrates for antiviral materials based on the chemical nature of metal elements, and promote the development of novel powerful weapons for virus elimination.

Metal nanoparticles not only have antiviral activity against viruses transmitted from person to person, but also have promising therapeutic effects on virus-infected animal and plant groups. In addition, considering the high cost and time-consuming process of new drugs discovery and the excellent physicochemical properties of metal nanoparticles, the proposal to use metal nanoparticles to inhibit the spread of viruses is valuable. Together, it is necessary to accelerate the research and development of metal nanoparticles, and more efforts need to be made to drive the translation of research results into actual industrialized products and devices. This will be beneficial to control the spread and infection of the virus, reduce the use of pesticides, and ultimately reduce the threat of the virus to human survival and environmental health.

## Figures and Tables

**Figure 1 nanomaterials-12-02345-f001:**
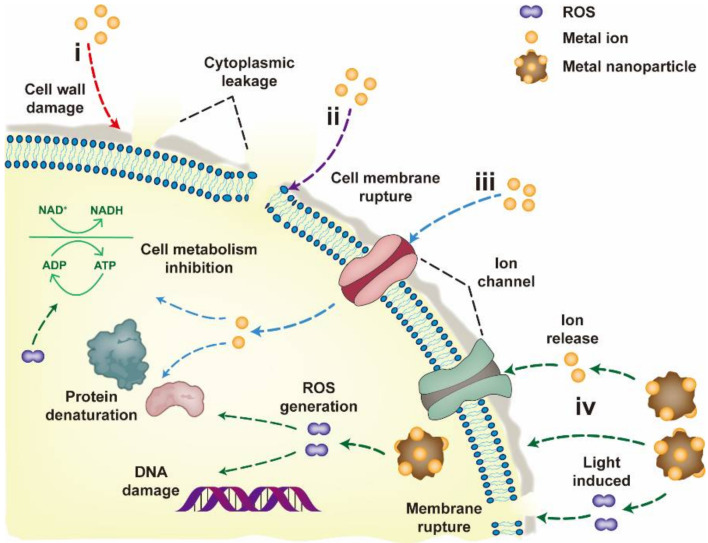
The main mechanisms of bacterial inactivation by metals. Different colored arrows are used to distinguish each mechanism.

**Figure 2 nanomaterials-12-02345-f002:**
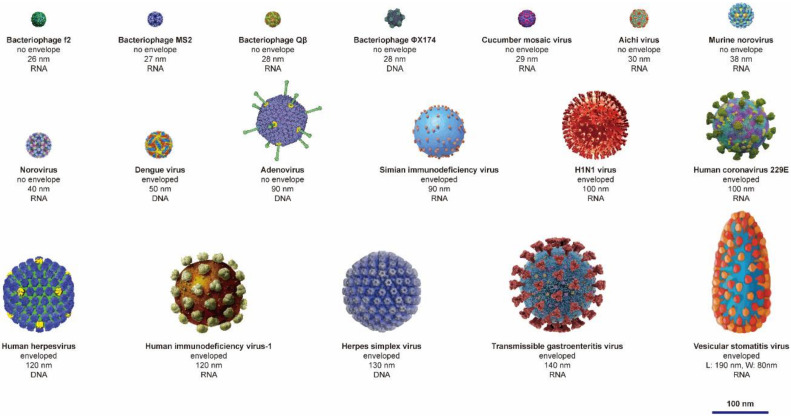
Schematic illustration of various viruses that can be inactivated by metallic materials. The annotations below each virus are the name, surface structure, diameter and genetic material of the virus. Color and cartoon are only used to facilitate understanding of the viral surface structure and do not fit perfectly with the true viral appearance.

**Figure 3 nanomaterials-12-02345-f003:**
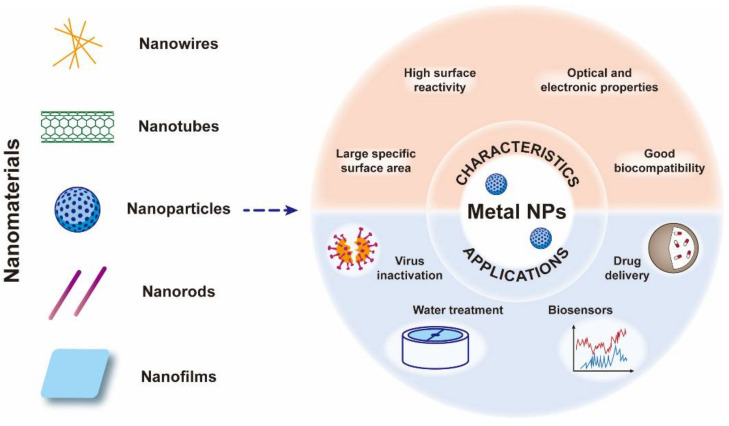
The types of nanomaterials and the advantages and applications of nanoparticles.

**Figure 4 nanomaterials-12-02345-f004:**
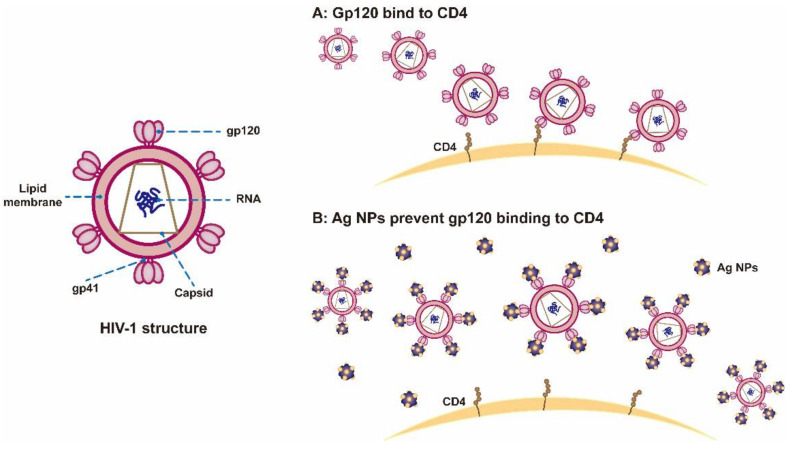
(**A**) The process of HIV-1 infecting host cells. (**B**) The mechanisms of HIV-1 inactivation by Ag NPs.

**Figure 5 nanomaterials-12-02345-f005:**
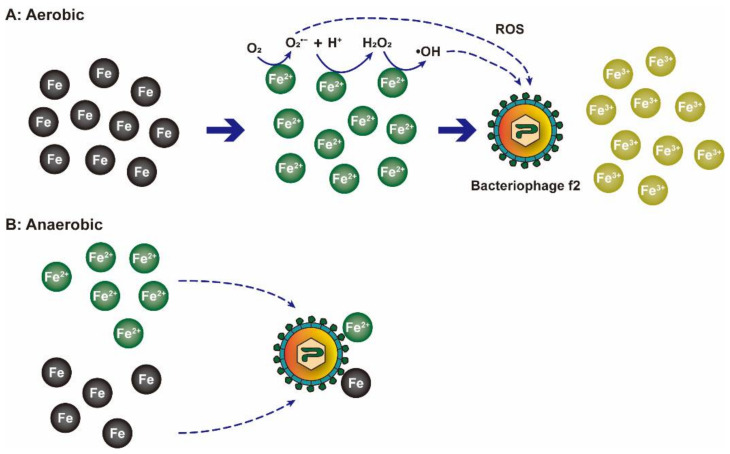
The mechanisms of bacteriophage f2 inactivation by NZVI in (**A**) aerobic and (**B**) anaerobic conditions. Adapted from Ref [24].

**Figure 6 nanomaterials-12-02345-f006:**
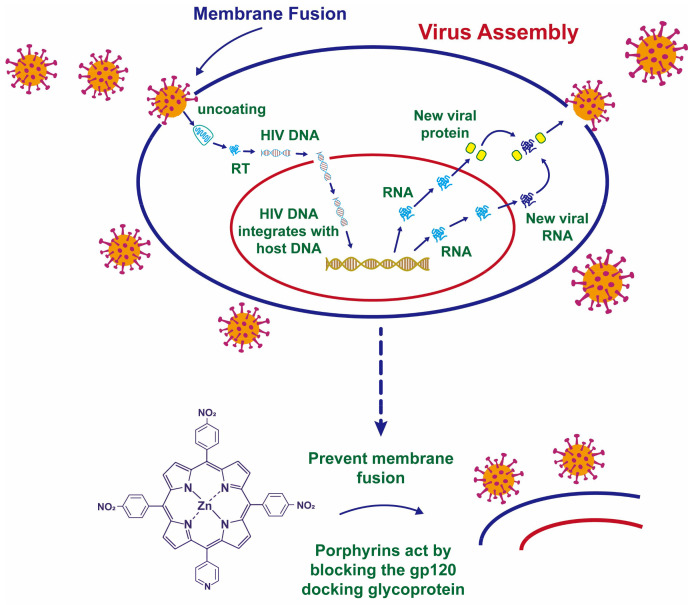
The process of SIVmac infecting host cells and the mechanisms of SIVmac inhibition by the nitroporphyrin-zinc complex. Adapted from Ref [52].

**Figure 7 nanomaterials-12-02345-f007:**
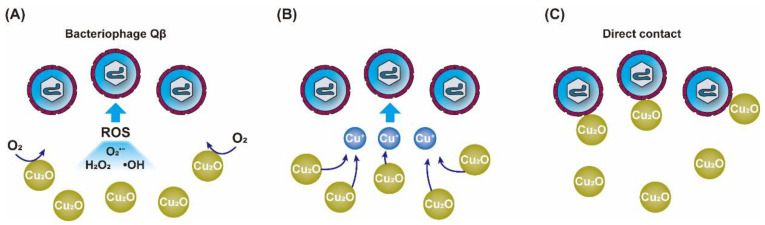
Three possible mechanisms of bacteriophage Qβ inactivation by Cu_2_O proposed by researchers. (**A**) Cu_2_O induces the generation of ROS to inactivate bacteriophage Qβ. (**B**) Cu_2_O releases Cu^+^ to inactivate bacteriophage Qβ. (**C**) Cu_2_O directly makes contact with bacteriophage Qβ for inactivation.

**Figure 8 nanomaterials-12-02345-f008:**
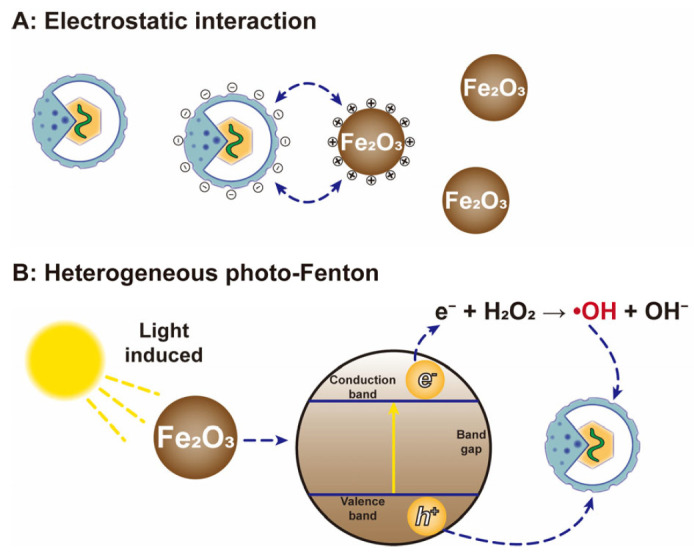
The mechanisms of bacteriophage MS2 inactivation by Fe_2_O_3_. (**A**) Electrostatic interactions between bacteriophage MS2 and Fe_2_O_3_ lead to bacteriophage MS2 inactivation. (**B**) ROS generated from heterogeneous photo-Fenton reactions leads to bacteriophage MS2 inactivation.

**Figure 9 nanomaterials-12-02345-f009:**
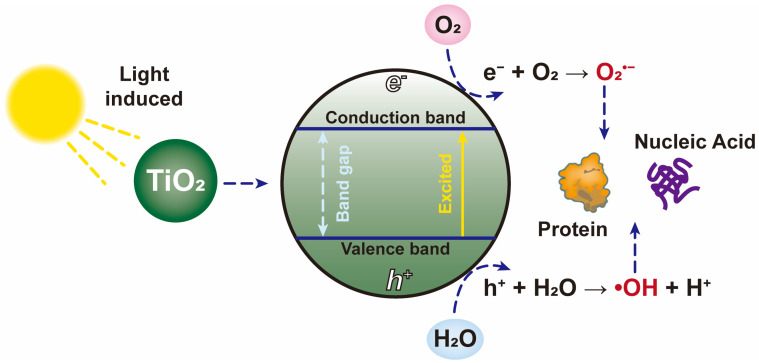
The mechanisms of virus inactivation by light-induced TiO_2_.

**Figure 10 nanomaterials-12-02345-f010:**
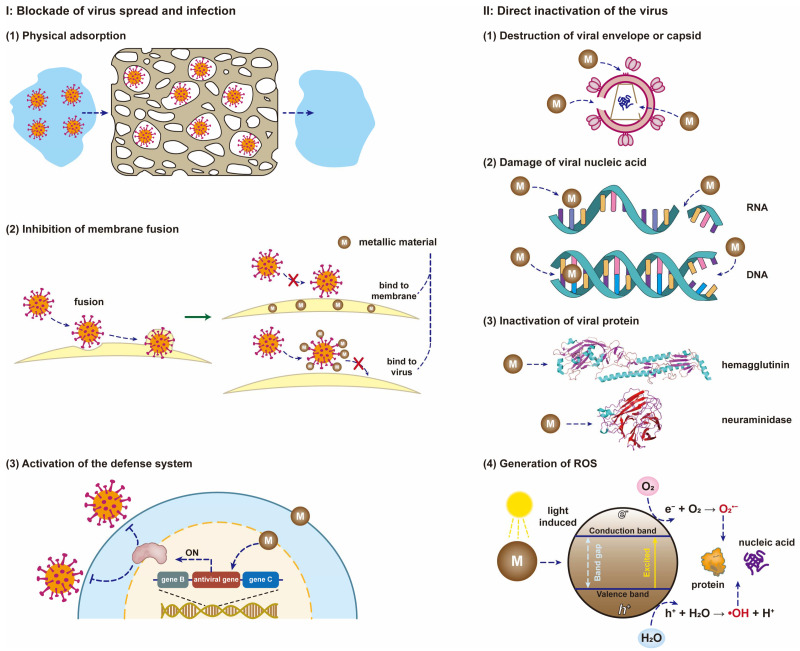
The antiviral mechanisms of metallic materials.

**Figure 11 nanomaterials-12-02345-f011:**
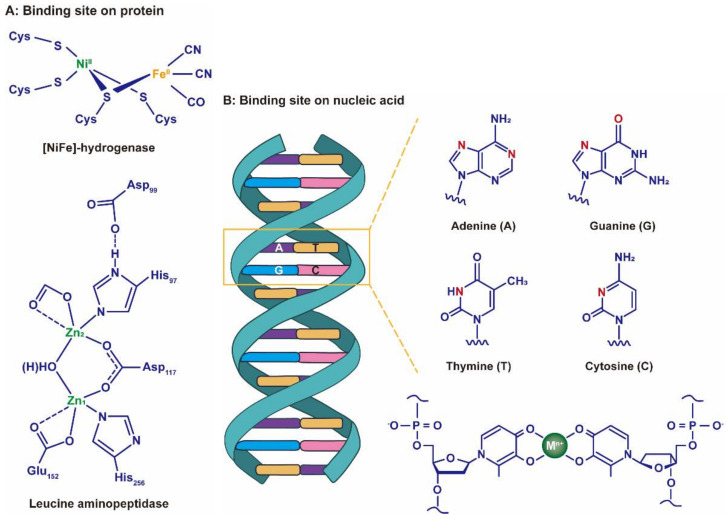
(**A**) The binding site on the protein. (**B**) The binding site on nucleic acid. Nitrogen and oxygen atoms highlighted in red are the predominant coordination sites for metal cations.

**Figure 12 nanomaterials-12-02345-f012:**
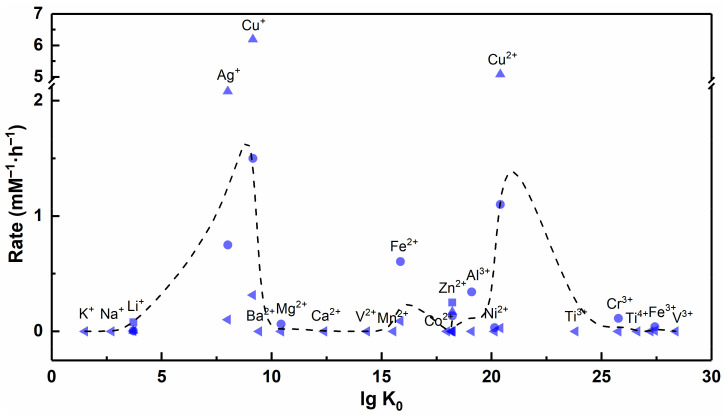
Relationship between the chelation equilibrium constant of the M^n+^ and antiviral rate. Multiple symbols for the same element represent results from different experiments. The dashed line is connected by the mean value of each element. The raw data are shown in Appendix A.

**Figure 13 nanomaterials-12-02345-f013:**
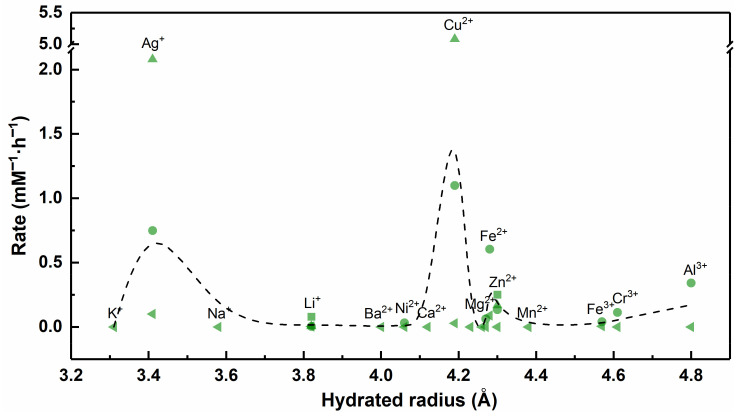
Relationship between the hydrated radius of the M^n+^ and antiviral rate. Multiple symbols for the same element represent results from different experiments. The dashed line is connected by the mean value of each element. The raw data are shown in Appendix A.

**Figure 14 nanomaterials-12-02345-f014:**
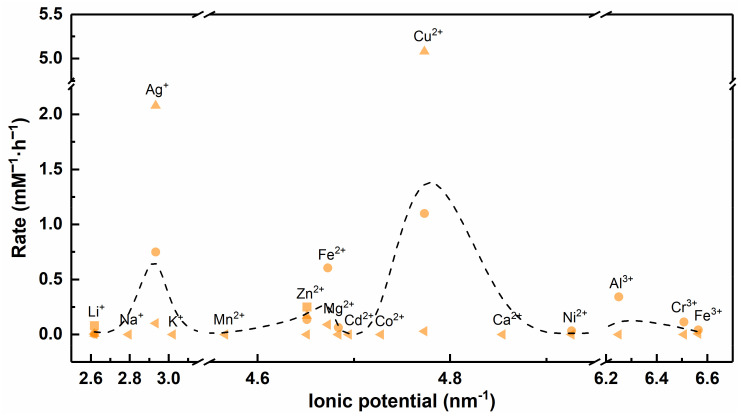
Relationship between the ionic potential of the M^n+^ and antiviral rate. Multiple symbols for the same element represent results from different experiments. The dashed line is connected by the mean value of each element. The raw data are shown in Appendix A.

**Figure 15 nanomaterials-12-02345-f015:**
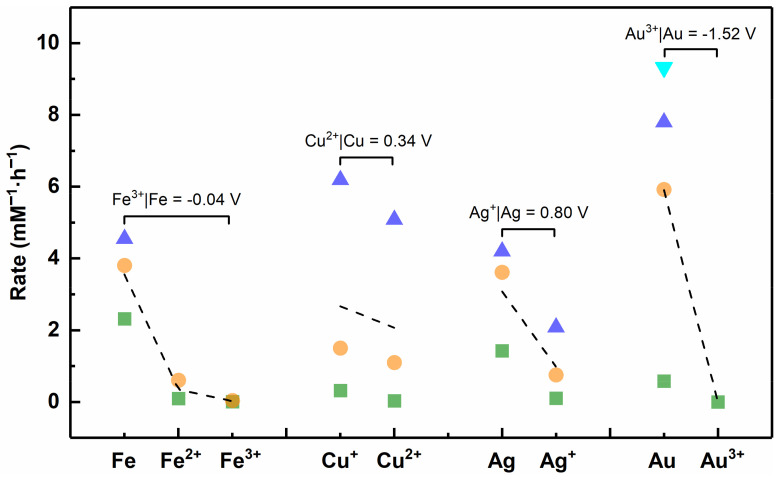
The antiviral rates of metal element in different valence states. Multiple symbols for the same element represent results from different experiments. The dashed line is connected by the mean value of each valence state. The raw data are shown in Appendix A.

**Table 1 nanomaterials-12-02345-t001:** Summary of the antiviral performances by metal nanoparticles.

Metallic Material	Size	Viruses	Mechanism	Reference
CuI NPs	D50 = 160 nm	H1N1 virus	Cu^+^ dissolved from NPs induced ROS production to destroy viral proteins (e.g., hemagglutinin and neuraminidase).	[20]
Cu_x_O_y_-Al_2_O_3_	30~50 nm	Bacteriophage MS2	Electrostatic adsorption, positively charged NPs bound negatively charged viruses.	[60]
Ag NPs	21 ± 18 nm	HIV-1 virus	The combination of Ag NPs and HIV-1 glycoprotein gp120.	[21]
Ag NPs	2~15 nm	SARS-CoV-2	Ag NPs damaged the surface proteins to affect the structural integrity of virions.	[64]
Ag NPs	5 nm	SARS-CoV-2		[65]
Ag30-SiO_2_ NPs	≈30 nm	Murine norovirus, Bacteriophage MS2	Ag^+^ dissolved from NPs bound to the thiol group of viral proteins. *^,a^	[50]
Ag NPs	27 ± 4 nm	HcoV-229E		[66]
NiO NPs	15~20 nm	Cucumber mosaic virus	NiO NPs activated the expression of defense-related genes in cells to resist CMV.	[22]
			Photocatalytic NiO NPs induced the production of ROS to destroy the virus structure. *	
Ni/Fe NPs	92.6 ± 3.5 nm	Bacteriophage f2	Ni as a catalyst for inactivation.	[71]
			Viruses were damaged by ROS which was generated during Fe oxidation.	
Au NPs	19~110 nm	Vesicular stomatitis virus	Au NPs attached to VSV and prevented VSV binding to host cells.	[23]
Au NPs	≈18.27 nm	Herpes simplex virus	Au NPs attached to the surface of HSV to eliminate the infectivity of the virus.	[73]
			Au NPs entered the host cells and interfered with viral replication.	
Au NPs	11 nm	Measles virus	High affinity between Au NPs and disulfide bonds prevented viral infection of host cells.	[76]
Au NPs	≈150 nm	Influenza virus	The disulfide bonds were cleaved by Au NPs to block membrane fusion.	[77]
NZVI	≈200 nm	Bacteriophage MS2	O_2_^•−^ played the major role in phase I and ·OH played the major role in phase II.	[81]
NZVI	<100 nm	Bacteriophage f2		[82]
NZVI	≈50 nm	Bacteriophage f2	NZVI were oxidized to Fe_3_O_4_ or Fe_2_O_3_ which adsorbed viruses in the initial stage.	[24]
			Fe^2+^ dissolved from NZVI generated ROS to inactivate viruses in the late stage.	

^a,^* There is investigation on the mechanism but no experimental verification.

**Table 2 nanomaterials-12-02345-t002:** Summary of the antiviral performances by metal ions.

Metal Ion	Source	Viruses	Mechanism	Reference
Cu^2+^	CuCl_2_, CuSO_4_	H9N2 virus	Cu^2+^ destroyed the structure of viruses.	[85]
Cu^2+^	CuZeo	H5N1 virus, H5N3 virus	Cu^2+^ destroyed the structure of viruses.	[53]
Ag^+^	AgNO_3_, Ag_2_O	Influenza A virus, bacteriophage Qβ	Ag^+^ broke disulfide and thiol bonds of viral proteins.	[88]
Ag^+^	Silver electrode	Sacbrood virus		[90]
Zn^2+^	ZnCl_2_, ZnSO_4_	Transmissible gastroenteritis virus	Zn^2+^ destroyed the RNA polymerase of viruses to shorten their life cycle. ^a,^*	[95]
Zn^2+^	Nitroporphyrin-zinc complexes	HIV-1 virus, SIVmac virus		[52]
Ag^+^, Cu^2+^, Zn^2+^	Hybrid coating	HIV-1 virus, H1N1 virus, Human herpesvirus, Dengue virus	Ag^+^ and Cu^2+^ ruptured the viral envelope or were bound to the thiol group of the viral proteins.	[97]

^a,^*: There is investigation on the mechanism but no experimental verification.

**Table 3 nanomaterials-12-02345-t003:** Summary of the antiviral performances by pure metals and alloys.

Metallic Material	Viruses	Mechanism	Reference
Copper	Adenovirus, Norovirus		[47]
Copper	Influenza A Virus	Copper directly damaged the RNA of viruses to prevent viral replication. ^a,^*	[51]
Copper, copper-nickel alloys, brass	Human coronavirus 229E	Cu^2+^ and Cu^+^ dissolved from copper alloy directly inactivated viruses.	[103]
		O_2_^•−^ generated on the surface of the alloy enhanced the inactivation effect.	
Zero-valent iron	Aichi virus, Adenovirus 41, Bacteriophage ΦX174 and MS2		[105]

^a,^*: There is investigation on the mechanism but no experimental verification.

**Table 4 nanomaterials-12-02345-t004:** Summary of the antiviral performances by metal compounds.

Metallic Material	Viruses	Mechanism	Reference
Cu_2_O	Influenza A virus, Bacteriophage Qβ	Cu_2_O damaged the function of viral hemagglutinin and neuraminidase.	[88]
Cu_2_O, Cu_2_S, CuI	Bacteriophage Qβ	Cuprous compounds adsorbed viral proteins.	[111]
Fe_2_O_3_	Bacteriophage MS2 and ΦX174	Fe_2_O_3_ adsorbed viruses.	[114]
		Light induced Fe_2_O_3_ to inactivate viruses.	
Fe_2_O_3_	Bacteriophage P22	Fe_2_O_3_ combined with viruses through electrostatic adsorption.	[115]
Fe_2_O_3_	Bacteriophage MS2	Fe_2_O_3_ directly bound to viruses through electrostatic interaction.	[116]
		Dissolved iron generated ROS which destroyed viral capsid or envelope.	
Cu-TiO_2_	Bacteriophage f2	Light induced Cu-TiO_2_ to generate ROS.	[49]
HA/TiO_2_	H1N1 virus	TiO_2_ was activated by UV lamps to produce ROS which destroyed the viral envelope, nucleic acids and proteins.	[121]

## Data Availability

Not applicable.

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
