# Peer review of "Chemical Nature of Metals and Metal-Based Materials in Inactivation of Viruses"

_nanomaterials, 2022, doi:10.3390/nano12142345_

Round 1

Reviewer 1 Report

The manuscript titled “Chemical nature of metal nanoparticles and other materials in inactivating of virus,” reviews the antiviral effects of metals and metal-based reagents against a selection of viruses, as well as discussing potential mechanisms of inactivation as well as challenges and future considerations. Overall, I think the review is interesting, well-organized, and will be of interest to the field; however, I have a number of comments that must be addressed before I can recommend this work for publication.

-Overall, I think the language could be improved a little, but I could understand what was being stated for the most part. However, I recommend changing the phrasing from “order of magnitude” to “logs” as it is more commonly used in viral inactivation work. Also, when discussing viral inactivation, it is important to state at every instance how it was measured (plaque assay—PFU/ml; TCID50, genomic copies via real time PCR, etc.) as this is important.

-I believe the authors should double check the proper nomenclature for viruses (herpesviruses is one word when discussing them as a group whereas herpes simplex virus is separated), though most viruses were okay (coxsackievirus B3 should be as written here); also it should be written E. coli, with a space between the genus and species, as well as italicized.

-Since more than just metal nanoparticles are discussed, I recommend changing the title to use the phrase “metals and metal-based materials” in the title instead of just metal nanoparticles. Also, use “viral inactivation” instead of the way it is written to be more grammatically correct.

-I would also make it clear that the review is not comprehensive in each of the sections, and that the studies discussed include a selection of work for each type of viral inactivation, rather than being comprehensive.

-I believe section 4 reads more like speculation rather than a review of observed results. I think the phrasing should be changed throughout to emphasize that this is speculative. For instance, “Potential Antiviral Mechanisms at the Physicochemical Level.” There also should be a more direct statement noting the lack of research that exists confirming these proposed mechanisms. Overall, it is quite general in language, even though viruses can dramatically differ in their physicochemical properties; thus, one could suspect the discussed mechanisms could be different between viruses. This point needs to be made more clearly (it is alluded to, and the section is interesting, but it reads more like a “Perspective” than a review).

-Lines 234-235: It should be noted that this type of test does not account for longer term toxicity that may bee seen in in vivo models like mice, as nanoparticles in particular can accumulate and cause damage over a less acute time than tissue culture.

-Line 341: I would double check the official designation with the ICTV, as I think this is just “sacbrood virus” and you should drop “Korean” from the name.

-Line 361: 1/25 is an odd way to report viral reduction, I would mention it in form of a percentage or log reduction (then state units to imply what method was used to measure it).

-Tables: In general, change the phrasing to mention mechanism was not studied in the cited work. Other work has investigated the mechanism for some of the virus/metals mentioned. For instance, copper alloys and norovirus in Table 3:

 (1) Warnes, S. L.; Keevil, C. W. Inactivation of Norovirus on Dry Copper Alloy Surfaces. PLoS One 2013, 8 (9), e75017.

(1)         Warnes, S. L.; Summersgill, E. N.; Keevil, C. W. Inactivation of Murine Norovirus on a Range of Copper Alloy Surfaces Is Accompanied by Loss of Capsid Integrity. Appl. Environ. Microbiol. 2015, 81 (3), 1085–1091.

(1) Manuel, C. S.; Moore, M. D.; Jaykus, L. A. Destruction of the Capsid and Genome of GII.4 Human Norovirus Occurs during Exposure to Metal Alloys Containing Copper. Appl. Environ. Microbiol. 2015, 81 (15), 4940–4946.

Also, for ionic copper:

(1) Mertens, B. S.; Moore, M. D.; Jaykus, L.; Velev, O. D. Efficacy and Mechanisms of Copper Ion-Catalyzed Inactivation of Human Norovirus. ACS Infect. Dis. 2022, 8 (4), 855–864.

-Tables: I also think it would be valuable to define abbreviations in the Tables, and in some cases write out what the acronyms stand for; for instance writing out hemagglutinin and neuraminidase in Table 4

-Line 534: Please describe what wuestite is composed of for the benefit of the reader.

-Line 551: It is unclear what “hole on the other hand” means—maybe just “on the other hand”?

-Lines 623-625: This is a little out of place and should be moved to where these proteins are discussed at first instance in the text.

-Figures in Section 4: Was this data generated from a simulation or calculation? The methods used to generate these are not provided. It is a little confusing as I think this further makes it read like a perspective rather than a review. Or was this from data from work that has been performed at the bench? This should be clarified.

-Lines 870-871: This statement is a little misleading, as metals and metal nanoparticles can have toxicity and many concerns still exist (the authors do a good job discussing these concerns in the next paragraph, lines 875-884).

-Line 863: I would mention humidity as well here as that can be quite important for the alloy/surface work.

-Lines 932-935: This point does not apply as much to viruses.

-Lines 942-949: It is a little unclear what point is being made here.

-Overall, I think this is a good review that will be of value to the field, but believe these comments need to be addressed first to improve it. 

Reviewer 2 Report

This review article's goal and scope are essential to the biomaterials community because they provide a complete understanding of the antiviral characteristics of metal nanoparticles, metal oxide, and metal-based compounds/alloys. In the broad spectrum of virus inactivation, the review paper summarizes and articulates the chemical nature of metal nanoparticles and other materials. This review paper delves into the antiviral mechanisms of metal nanoparticles and other compounds. The illustrations are quite helpful in demonstrating the antiviral mechanism. I have a couple of comments for the authors to address in their revised manuscript:

1. When describing the antiviral properties of Ag NPs, it would be desirable to summarize and expand on the size-dependent antiviral properties in general, as well as cite more publications on the antiviral properties of various Ag NP sizes.

2. The authors need to replace lines 49 and 50 a slightly modified version to reflect better structure and up-to-date referencing:

"Considering the time consuming process of new drug discovery and short comings of conventional techniques, many researchers have been committed to seeking effective, simple and accessible methods to inhibit virus spread, including artificial intelligence approaches [ Antiviral nanoparticle ligands identified with datamining and high-throughput virtual screening, EP Booker et al., RSC Advances 11 (37), 23136-23143 ].

3. When talking about protective equipment ( Cu- Zeo-based protective equipment on line 318). It would be instructive to do the same for the Ag ion discussion in the section following the copper one. The authors are encouraged to add after line 351 a sentence to the effect of the use of Ag nanoparticles in personal protective equipment (such as face mask) against SARS-Cov2 virus, the cause of COVID-19 illness and support with a reference [example:  M. Abulikenmu et al. "Silver Nanoparticle-Decorated Personal Protective Equipment for Inhibiting Human Coronavirus Infectivity", ACS Appl. Nano Mater. 2022, 5, 1, 309–317

Reviewer 3 Report

The review paper includes plenty of information regarding metal nanoparticles and possible interactions and inhibition mechanisms towards viral activity.

In this sense, the organization of the manuscript results confusing for me. It starts listing different types of materials, then the corresponding tables (in such format, layout and format that they seem to be cut). Later, one can find a description of inhibition mechanisms which I consider is the added value aspect of the review. I would include in detail the inhibition mechanisms and then mention the examples. Not the way they are presented right now. In addition, lack of reference used for the images – figures (unless they are property of the authors.

I do not get the term “technique” in line 16.

English must be checked through the whole text since there are a lot of long phrases which result confusing for the reader.

I recommend major revisions

Round 2

Reviewer 1 Report

I thank the authors for thoughtfully and comprehensively addressing my comments. I think there are still occasional awkward phrases so another read-through for English may improve it, but from a content standpoint, all of my comments have been very well addressed--this will serve as a very valuable contribution to the field that is quite in depth and interesting. 

Author Response

Dear Editor and Reviewers,

We thank you very much for your comments and suggestions in regards to our manuscript entitled “Chemical nature of metal nanoparticles and other materials in inactivating of virus (ID: 1763407). The comments and suggestions have all been valuable in both improving this manuscript, as well as guiding us in future studies. We have reviewed each comment carefully and made corresponding corrections which we hope will address the reviewers’ concerns. Below, please find our detailed response to each of the reviewers’ comments.

(Reviewers’ comments are shown in black, our responses are shown in blue, and the revisions in the manuscript are in red. The line numbers refer to those in the revised manuscript, not the clean version.)

Reviewer #1

Comments:

I thank the authors for thoughtfully and comprehensively addressing my comments. I think there are still occasional awkward phrases so another read-through for English may improve it, but from a content standpoint, all of my comments have been very well addressed--this will serve as a very valuable contribution to the field that is quite in depth and interesting.

Major:

  1. I think there are still occasional awkward phrases so another read-through for English may improve it.

Response: Thank you very much for your comments. We have rechecked this article and revised the incorrect description.

Reviewer 3 Report

Thank you for submitting the revised version of the manuscript. Implemented changes have greatly improved the quality and the understanding of it. 

I am still concerned regarding the copyright of the figures. For instance, in figure 3, it seems you used Biorender to design some of the particles included there. You should have a license of this software and state this in your manuscript if you want the figure to be used in public and publishable material. So please, have a look at that and confirm you have the permissions for all figures. 

Author Response

Dear Editor and Reviewers,

We thank you very much for your comments and suggestions in regards to our manuscript entitled “Chemical nature of metal nanoparticles and other materials in inactivating of virus (ID: 1763407). The comments and suggestions have all been valuable in both improving this manuscript, as well as guiding us in future studies. We have reviewed each comment carefully and made corresponding corrections which we hope will address the reviewers’ concerns. Below, please find our detailed response to each of the reviewers’ comments.

(Reviewers’ comments are shown in black, our responses are shown in blue, and the revisions in the manuscript are in red. The line numbers refer to those in the revised manuscript, not the clean version.)

Reviewer #3

Comments:

Thank you for submitting the revised version of the manuscript. Implemented changes have greatly improved the quality and the understanding of it.

Major:

  1. I am still concerned regarding the copyright of the figures. For instance, in figure 3, it seems you used Biorender to design some of the particles included there. You should have a license of this software and state this in your manuscript if you want the figure to be used in public and publishable material. So please, have a look at that and confirm you have the permissions for all figures.

Response: Thank you very much for your comments. In this article, all figures were created with Adobe Illustrator 2020. We have mentioned the drawing software in line 87-88.

Line 87-88: …of bacteria and hinders bacterial propagation (Figure 1, all figures were created with Adobe Illustrator 2020)…